# On the quality of commercial chemical vapour deposited hexagonal boron nitride

Yue Yuan[1], Jonas Weber [1], Junzhu Li [1], Bo Tian [1], Yinchang Ma [1], Xixiang Zhang [1], Takashi Taniguchi [2], Kenji Watanabe[3] & Mario Lanza [1] ✉

The semiconductors industry has put its eyes on two-dimensional (2D) materials produced by chemical vapour deposition (CVD) because they can be grown at the wafer level with small thickness fluctuations, which is necessary to build electronic devices and circuits. However, CVD-grown 2D materials can contain significant amounts of lattice distortions, which degrades the performance at the device level and increases device-to-device variability. Here we statistically analyse the quality of commercially available CVD-grown hexagonal boron nitride (h-BN) from the most popular suppliers. h-BN is of strategic importance because it is one of the few insulating 2D materials, and can be used as anti-scattering substrate and gate dielectric. We find that the leakage current and electrical homogeneity of all commercially available CVD h-BN samples are significantly worse than those of mechanically exfoliated h-BN of similar thickness. Moreover, in most cases the properties of the CVD h-BN samples analysed don't match the technical specifications given by the suppliers, and the sample-to-sample variability is unsuitable for the reproducible fabrication of capacitors, transistors or memristors in different batches. In the short term, suppliers should try to provide accurate sample specifications matching the properties of the commercialized materials, and researchers should keep such inaccuracies in mind; and in the middle term suppliers should try to reduce the density of defects to enable the fabrication of high-performance devices with high reliability and reproducibility.

Two-dimensional (2D) layered materials possess extraordinary physical, chemical, electronic, thermal, and optical properties, which are used in a wide range of applications[1]. The first commercial applications of 2D materials used liquid phase exfoliated (LPE) samples, which consist of a solution containing flakes of a specific 2D material that can be deposited on arbitrary substrates by various methods (e.g., spinning, spraying, inkjet printing). These LPE 2D materials can be used as a coating to provide conductivity and/or mechanical strength to different objects[1] or to pattern electrical circuits (among others)[2,3]. In 2019, a study[4] revealed that most of the commercially available LPE 2D materials were characterized by very poor quality (the lateral size of the flakes was small, their thickness was high, and often contained aggregates and large powder particles)—in contrast to the advertised specifications on the website of the suppliers. This article changed the communities' perspective on which precautions to take when purchasing LPE 2D materials[5], helped the researchers to better understand the reliability of reported findings, and prompted many companies to improve the quality of their samples and/or to report more accurate product information on their websites.

[1]Materials Science and Engineering Program, Physical Science and Engineering Division, King Abdullah University of Science and Technology (KAUST), Thuwal 23955-6900, Saudi Arabia. [2]International Center for Materials Nanoarchitectonics, National Institute for Materials Science, 1-1 Namiki, Tsukuba 305-0044, Japan. [3]Research Center for Functional Materials, National Institute for Materials Science, 1-1 Namiki, Tsukuba 305-0044, Japan. ✉e-mail: mario.lanza@kaust.edu.sa

Now, the semiconductors industry has put its eyes on 2D materials produced by chemical vapour deposition (CVD) because this method provides controllable thickness and relatively small thickness fluctuations on industrial wafers (300 mm in diameter)[6,7], and interesting prototypes have been reported[8,9]. However, commercially available CVD-grown 2D materials contain local defects, such as lattice distortions and impurities, and the density of defects increases with the thickness. While few groups have managed to produce monolayer 2D materials of relatively high quality via CVD[10,11], their samples are not accessible to everyone, and most scientists employ commercially available CVD-grown 2D materials for their studies and prototype devices. Moreover, the production of high-quality multilayer 2D materials via the CVD method remains challenging. Hence, revising the quality of the commercially available CVD-grown 2D materials is very important.

In this article, we evaluate the morphological and electronic properties of commercially available hexagonal boron nitride (h-BN) grown by the CVD method. This 2D material is of strategic importance for the semiconductor industry given its high band gap of ~5.9 eV[12]—it is one of the few insulating 2D materials—which allows its use as dielectric in capacitors[13], transistors[14], and memristors[15,16]. We analyse CVD-grown h-BN samples from the most popular suppliers using cross-sectional transmission electron microscopy (TEM), conductive atomic force microscopy (CAFM), scanning electron microscopy (SEM), and Raman spectroscopy. The results are compared with mechanically exfoliated h-BN stacks and industrial quality $SiO_2$ of similar thickness, which serve as a reference. The main conclusion of our study is that the leakage current and electrical homogeneity of commercially available CVD h-BN samples are significantly worse than those of mechanically exfoliated h-BN. Moreover, in many cases, the properties of the CVD h-BN samples analysed do not match the technical specifications given by the suppliers, and the sample-to-sample variability is found to be unsuitable for the reproducible fabrication of electronic devices in different batches.

## Results

### Mechanically exfoliated monolayer h-BN

We start by analysing a reference sample that consists of mechanically exfoliated monolayer h-BN flakes placed on a conductive substrate. Previous studies employed $SiO_2$/Si substrates coated with thin metallic films (such as 50 nm Au) using sputtering or electron beam evaporator[17]. However, the typical surface roughness of the metallic films deposited with such methods is relatively high (above 0.5 nm), which produces the apparition of unwanted gaps between the 2D material and the substrate, leading to false nanoscale electrical characterization (see Supplementary Fig. 1). To avoid this problem, in our investigation, we use ultra-flat substrates that consist of 5 nm Ru/ 30 nm Ta/300 nm $SiO_2$/Si (from top to down), which have a root mean square (RMS) surface roughness of 0.310 nm (see Supplementary Fig. 2a). The Ru film is highly conductive because current maps without applying any bias show high currents above 12 pA everywhere (see Supplementary Fig. 2b). These currents are produced by the inherent offset voltage of the CAFM[18,19], which in our machine is around 7.25 mV. We also determine that the contact resistance of our setup is 500 kΩ, as a voltage difference of 375 μV produces a current increase of 750 pA (see Supplementary Fig. 2c).

We then mechanically exfoliate monolayer h-BN flakes and deposit them on the Ru/Ta/$SiO_2$/Si substrate (see Fig. 1a, b). The RMS roughness of the h-BN surface is 0.273 nm, which is only slightly lower than that of the Ru substrate. This demonstrates that the h-BN adheres well to the surface of the Ru substrate and that no significant gaps are formed. The thickness of the h-BN flakes analysed ranged between 0.26 and 0.44 nm (see Fig. 1c and Supplementary Fig. 3), with an average thickness of 0.367 nm, very similar to that measured in other experimental studies[20] and theoretically calculated via ab initio

simulations[21]. Next, we apply ramped voltage stresses (RVS) at 24 randomly selected locations of the surface of the mechanically exfoliated monolayer h-BN flake on the Ru substrate (using a current limitation of 100 pA to avoid CAFM tip degradation) and create a current versus voltage (I–V) plot. As Fig. 1f shows, at all locations, the currents increase rapidly, and the minimum voltage needed to observe current above the noise level (~3 pA), called $V_{ON}$, is low (0.21 ± 0.08 V); the resistance of the last data point measured before reaching the current limitation is 11.73 ± 5.50 GΩ. The point-to-point variability of the I–V plot is quite low, even lower than that observed (using the same setup) in other high-quality and homogeneous ultra-thin insulators, such as industrial-quality $SiO_2$ (see Supplementary Fig. 4). These small variations may be related to small thickness fluctuations of the van der Waals gap between the h-BN and the Ru substrate. The same experiment is repeated on different h-BN flakes with different thicknesses: 1.35 nm (~4 layers), 2 nm (~6 layers), 3.2 nm (~10 layers), 5.3 nm (~16 layers), and 5.65 nm (~17 layers), as shown in see Supplementary Fig. 5. In all cases the I–V curves group together around a mean $V_{ON}$, although the variability of $V_{ON}$ from one location to another within each sample is larger than for monolayers.

We collect current maps without applying any bias in the mechanically exfoliated monolayer samples, and no current is observed (Fig. 1g), meaning that the sample is mainly free of pinholes. Some locations next to the edge showed some pinholes (see Supplementary Fig. 6), probably due to the higher mechanical stress during peeling, but such behaviour is not representative of the entire surface of the monolayer h-BN flake. When a pinhole-free region of the monolayer h-BN flake is scanned under 0.8 V, we observe the presence of some local conductive spots with typical diameters of 3.24 ± 3.21 nm, which drive maximum currents of 20.2 pA (Fig. 1h). Considering that the mechanically exfoliated h-BN is free of defects (as demonstrated in several studies[17]), these local higher currents could only be explained by a small reduction of the van der Waals gap. If the same area is scanned at 0.8 V for four times, the size and the currents driven by the spots slightly increase (see Supplementary Fig. 7b–e), indicating a progressive degradation of the h-BN film. We select four current spots that appear in all scans and plot the resistance in each scan for each spot (see Fig. 1i). However, the damage to the h-BN stack is not very significant because subsequent scans without bias do not show remarkable currents above the noise level and no surface modification is observed (see Supplementary Fig. 7f).

In the literature, we only found two articles reporting CAFM data of ultra-thin (<1 nm) mechanically exfoliated h-BN, probably due to the difficulty of finding such thin flakes under the optical microscope of the CAFM. The first one presents I–V curves measured on monolayer, bilayer and trilayer h-BN on an Au-coated mica substrate, and it shows currents of 100 pA at voltages of 0.1, 0.25 and 0.45 V (respectively), which equals to contact resistances of 1, 2.5 and 4.5 GΩ[22]. Unfortunately, the authors only presented one I–V curve per sample (i.e., thickness), did not characterize point-to-point variability, and did not analyse the properties of the current spots in lateral maps. The second study presents current maps of bilayer and tetralayer h-BN on a graphite substrate and observed resistances of 10 MΩ and 1 GΩ, respectively[23]. While in that study, the sample appears to be homogeneous, the point-to-point variability within each layer may be masked by the huge logarithmic resistance scale used (which expands from 10 kΩ to 1 GΩ). The large difference in the resistances observed in these two studies (for bilayers, 2.5 GΩ[22] versus 10 MΩ[23]) may be related to the different substrates and/or to other CAFM experimental factors like tip/sample contact force, tip radius and metal coating, and relative humidity, or due to the different voltages at which the resistance was read (in ref. 23 this is not specified). In fact, comparing absolute current values from one CAFM study to another may be misleading if identical experimental conditions were not used. In Fig. 1 and Supplementary Figs. 5, 6, we provide additional insights and valuable

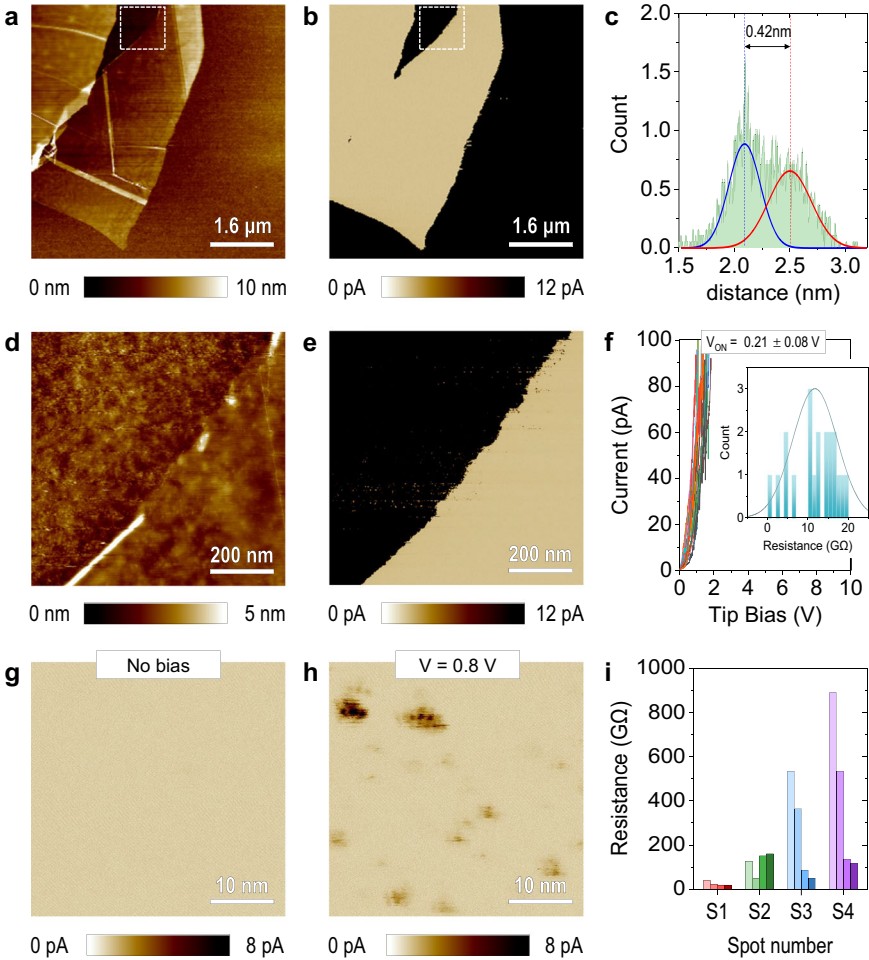

**Fig. 1 | Characterization of mechanically exfoliated monolayer hexagonal boron nitride (h-BN). a, b** and **d, e** Conductive atomic force microscopy (CAFM) topography maps and current maps collected without applying bias on a monolayer mechanically exfoliated h-BN flake transferred on a 5 nm Ru/30 nm Ta/ 300 nm $SiO_2$/Si substrate. The Ru film has been connected to the metallic sample holder using Ag ink to allow electrical measurements. **d** and **e** Are the zoom-in images marked as the white squares in (**a** and **b**). **c** Thickness analysis of the exfoliated h-BN flake in (**d**). The solid blue and red curves in **c** represent the Gaussian fittings for the original curves (in the colour of green), while the dashed blue and red lines mark the peak locations of each Gaussian distribution curve. **f** I–V curves collected at 24 randomly selected locations of the surface of the mechanically exfoliated monolayer h-BN flake on Ru substrate, in (**a**). The value of onset potential ($V_{ON}$) for each I–V curve and the resistance (read right before current saturation) are also shown, where the $V_{ON}$ is defined as the minimum voltage detected when the current just exceeds the noise level (~3 pA). **g** and **h** are the high-resolution current maps inside the mechanically exfoliated monolayer h-BN flake collected with biases of 0 and 0.8 V, respectively. **i** Evolution of the resistance of 4 current spots in 4 consecutive current maps (see Supplementary Fig. 7). For each spot, the left, middle-left, middle-right and right columns indicate the resistance detected in the 1st, 2nd, 3rd and 4th scan, respectively.

information about the homogeneity of mechanically exfoliated monolayer h-BN, determining how the current across the weakest locations evolves as the electrical stress proceeds.

## CVD-grown h-BN samples specifications

Now that the (defect-free) mechanically exfoliated monolayer h-BN has been statistically characterized and its electrical properties are well understood, we can characterize the commercially available CVD-grown h-BN from different suppliers and extract conclusions reliably. We purchased 19 CVD-grown h-BN samples from 9 different suppliers, namely Suppliers 1–9. These suppliers have been widely employed in multiple publications, many of them being highly cited (see Fig. 2). More specifically, we purchased 1 CVD-grown monolayer h-BN samples from each one of the 9 suppliers, and 10 CVD-grown multilayer h-BN samples from 6 different suppliers (5 samples from Supplier 1, and 1 sample from each of Suppliers 4–7, and 9). The total cost of all these samples together was US$7342. Table 1 shows a list of all the samples purchased, including some key specifications provided by the manufacturers: the percentage of the substrate that is covered by h-BN

(namely, coverage) and the thickness of the h-BN. The suppliers refer to these values as nominal or average, and, to the best of our knowledge, none of them provide publicly available information about statistical variability from one point to another in the same sample or from one sample to another. For monolayer samples, Suppliers 5–8 do not appear to provide information about coverage on both their websites and the sample datasheets; for multilayer samples, Suppliers 5–7 did not provide any thickness, just labelled the samples as "multilayer" on both their websites and the sample datasheets. We note that the missing information can significantly hamper experimental design and material applications.

## CVD-grown commercial h-BN labelled as monolayer

First, we analyse all the h-BN samples labelled as monolayers from suppliers, grown by CVD method on Cu foil using an optical microscope, after being transferred on a flat $SiO_2$/Si (surface roughness 200 pm) for better contrast. The images reveal that all the CVD-grown samples are continuous, and no appreciable thickness fluctuations can be observed (see Supplementary Fig. 8). Next, we collect cross-

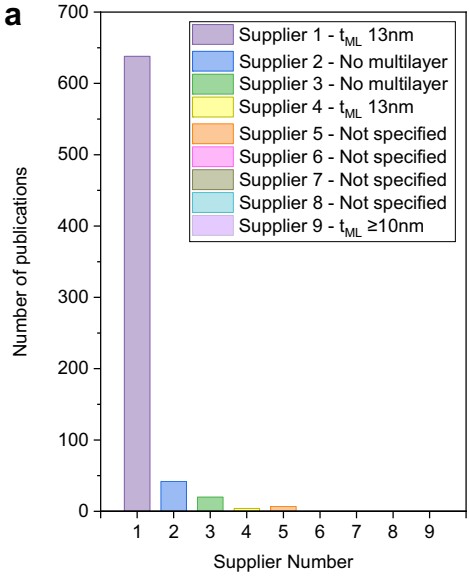
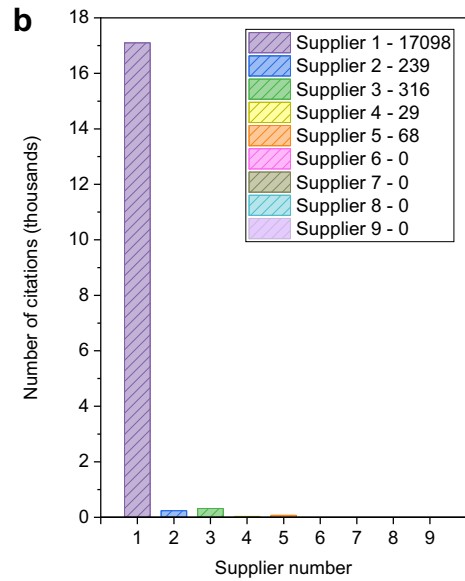

**Fig. 2 | Articles using commercially available chemical vapour deposition (CVD) h-BN. a** Number of publications using CVD h-BN from 9 different suppliers. **b** number of times that the articles in panel **a** have been cited. The information was collected by using Google Scholar, searching the name of supplier+"boron nitride" (on May 21, 2023). We detected 711 articles, 485 of them with at least one citation. All the articles together were cited 17,750 times. We also downloaded and inspected one by one all the articles cited 30 or more times (in total 46 articles), which altogether have been cited 6318 times. We confirm that in these most cited articles, the name of the supplier appeared as the provider of the h-BN for the experiments. Note that these numbers do not include articles that did not explicitly write the name of the supplier, meaning that the real use of commercially available CVD h-BN could be even higher. This might also explain the absence of articles mentioning Suppliers 6–9. $t_{ML}$ indicates the thickness of the multilayer stack specified by the manufacturers.

sectional TEM, top-view SEM, Raman spectroscopy as well as topographic/current maps and RVS via CAFM. The data collected with the CVD h-BN from Suppliers 1 and 2 (the most widely used, see Fig. 2) are shown in Fig. 3, and the data for Suppliers 3–9 are shown in Supplementary Fig. 9.

We analysed the first sample (from Supplier 1) by collecting 25 consecutive cross-sectional TEM images and observed that only 80% were 2D layered, although with a thickness ranging from 2 to 2.3 nm, and the remaining 20% contained a high density of defects, mainly erratic atomic bonding (see Fig. 3a, b and Supplementary Fig. 10). Sometimes these defects consist of one/few atoms and are observed in the TEM images as a small discontinuity of one/few layers, or as interstitial atoms between layers (see red arrows in Supplementary Fig. 10). But in other cases, this defective bonding propagates laterally and vertically over larger areas, which creates heavily disordered quasi-amorphous regions (see yellow arrows in Supplementary Fig. 10). Note that these local defects are not related to amorphization produced by the focused ion beam (FIB), as when the same experiments are carried out in mechanically exfoliated samples, a perfect layered structure is observed (see Supplementary Fig. 11). SEM image of the sample shows the presence of long wrinkles (see Fig. 3c), which indicates that the h-BN sheet is continuous, and several particles on the surface of the sample can also be observed.

When analysing the same sample with the CAFM, the RMS surface roughness appears to be high (8.96 nm, see Fig. 3d), but that is related to the morphology of the Cu foil below the h-BN sheet (see Supplementary Fig. 12), and it does not affect the electrical properties of the h-BN stack—the leakage current depends on the thickness and number of defects[24]. Current maps collected on the surface of the sample without applying any bias (Fig. 3e) display insulating grains with an average size of $0.54 \pm 0.38\ \mu m^2$ (red dot) that are separated by conductive grain boundaries (blue dot). By off-line processing Fig. 3e, we identify that the percentages of the area covered by the insulating grains and conductive grain boundaries are 82.27% and 17.73%, respectively, which is very similar to the length of layered regions

versus defective regions observed via cross-sectional TEM (80% versus 20%). The presence of long wrinkles in the SEM image (Fig. 3c) and the lack of a granular pattern in the topographic map (Fig. 3d) indicate that the h-BN sheet is continuous from a morphological point of view but discontinuous from a dielectric point of view (Fig. 3e). This should be related to the presence of many native defects in the CVD-grown h-BN, which appear to be heavily disordered (quasi-amorphous) in the cross-sectional TEM images (Fig. 3b).

Next, we applied RVS at 100 different locations on the surface of this CVD h-BN sample that is labelled as monolayer (which, as discussed, is, in fact, 2-nm-thick and discontinuous; see Fig. 3a–e). As Fig. 3f shows, most of the $I–V$ curves deviate from the typical ones measured on the mechanically exfoliated monolayer h-BN sample (Fig. 1f): (i) at 20 locations, the currents increase very fast and linearly (even faster than across the mechanically exfoliated monolayer sample), which should be related to the presence of pinholes and/or defective bonding and (ii) at 26 locations $V_{ON}$ is higher than 1 V, probably because most of the h-BN is ~2 nm thick although the insulating contribution of the randomly formed wrinkles may have further reduced the tunnelling current at few locations. We repeated the RVS experiments on a piece of the same h-BN but transferred on the same type of Ru substrate used for the mechanically exfoliated sample, and similar trends were observed (see Supplementary Fig. 13), probably due to the similar work function of Cu (4.52 eV) and Ru (4.71 eV). The fact that all the $I–V$ curves reach high currents of 100 pA at voltages ranging from 0.025 to 4 V, which equals contact resistances from 250 MΩ to 40 GΩ, indicates that the surface of the sample is free of contaminants (no polymer residues from the transfer). Moreover, we transferred the CVD h-BN and the mechanically exfoliated h-BN onto a 300 nm $SiO_2$/Si substrate for Raman spectroscopy characterization (see Supplementary Fig. 14) and compared it to the Raman signal obtained in mechanically exfoliated samples. We observed that: (i) the mechanically exfoliated sample shows homogeneous and strong $E_{2g}$ band of h-BN at 1367.02 $cm^{-1}$ with a full width at half maximum (FWHM) of 12.8 $cm^{-1}$; and (ii) the CVD h-BN from Supplier 1 shows very

**Table 1 | Summary of the properties of each commercially available chemical vapour deposition (CVD)-grown hexagonal boron nitride (h-BN) samples**

| | | | Coverage | | Thickness | | Onset potential | |
|---|---|---|---|---|---|---|---|---|
| | | | Value | Method | Value | Method | Value | Method |
| Monolayer | Supplier 1 | Specification | 90–95% | Not specified | Monolayer | Not specified | $0.21 \pm 0.08$ V * | CAFM |
| | | Measured | ~80% | CAFM, TEM | ~2 nm | TEM | $0.78 \pm 0.88$ V | CAFM |
| | Supplier 2 | Specification | 100% | Not specified | Monolayer | Not specified | $0.21 \pm 0.08$ V * | CAFM |
| | | Measured | 100% | CAFM, TEM | 0.33–1.32 nm | TEM | $4.20 \pm 1.23$ V | CAFM |
| | Supplier 3 | Specification | 100% | Not specified | Monolayer | Not specified | $0.21 \pm 0.08$ V* | CAFM |
| | | Measured | 100% | CAFM, SEM | Not measured | – | $2.47 \pm 0.88$ V | CAFM |
| | Supplier 4 | Specification | 90–95% | Not specified | Monolayer | Not specified | $0.21 \pm 0.08$ V* | CAFM |
| | | Measured | 100% | CAFM, SEM | Not measured | – | $2.56 \pm 1.35$ V | CAFM |
| | Supplier 5 | Specification | Not specified | – | Monolayer | Not specified | $0.21 \pm 0.08$ V * | CAFM |
| | | Measured | 100% | CAFM, SEM | Not measured | – | $2.33 \pm 0.81$ V | CAFM |
| | Supplier 6 | Specification | Not specified | – | Monolayer | Not specified | $0.21 \pm 0.08$ V * | CAFM |
| | | Measured | 100% | CAFM, SEM | Not measured | – | $6.94 \pm 1.33$ V | CAFM |
| | Supplier 7 | Specification | Not specified | – | Monolayer | Not specified | $0.21 \pm 0.08$ V * | CAFM |
| | | Measured | 100% | CAFM, SEM | Not measured | – | $4.05 \pm 0.81$ V | CAFM |
| | Supplier 8 | Specification | Not specified | – | Monolayer | Not specified | $0.21 \pm 0.08$ V* | CAFM |
| | | Measured | 100% | CAFM, SEM | Not measured | – | $1.51 \pm 1.02$ V | CAFM |
| | Supplier 9 | Specification | ~100 % | Not specified | Monolayer | Not specified | $0.21 \pm 0.08$ V * | CAFM |
| | | Measured | 100% | CAFM, SEM | Not measured | – | $3.25 \pm 0.91$ V | CAFM |
| Multilayer | Supplier 1 *Sample 1* | Specification | Not specified | – | Average 13 nm | AFM | 12 V* | CAFM |
| | | Measured | 100% | CAFM, TEM | ~1.7 nm | TEM | $1.63 \pm 1.02$ V | CAFM |
| | Supplier 1 *Sample 2* | Specification | Not specified | – | Average 13 nm | AFM | 12 V * | CAFM |
| | | Measured | 100% | CAFM, TEM | ~4.0 nm | TEM | $2.68 \pm 1.42$ V | CAFM |
| | Supplier 1 *Sample 3* | Specification | Not specified | – | Average 13 nm | AFM | 12 V* | CAFM |
| | | Measured | 100% | CAFM, TEM | ~5.0 nm | TEM | $4.25 \pm 0.96$ V | CAFM |
| | Supplier 1 *Sample 4* | Specification | Not specified | – | Average 13 nm | AFM | 12 V* | CAFM |
| | | Measured | 100% | CAFM, TEM | ~7.0 nm | TEM | $5.12 \pm 1.37$ V | CAFM |
| | Supplier 1 *Sample 5* | Specification | Not specified | – | Average 13 nm | AFM | 12 V* | CAFM |
| | | Measured | 100% | CAFM, TEM | ~8.3 m | TEM | $6.09 \pm 0.68$ V | CAFM |
| | Supplier 4 | Specification | Not specified | – | 13 nm | Not specified | 12 V * | CAFM |
| | | Measured | 100% | CAFM, SEM | Not measured | – | $5.34 \pm 0.47$ V | CAFM |
| | Supplier 5 | Specification | Not specified | – | Not specified | – | Unknown | – |
| | | Measured | 100% | CAFM, SEM | Not measured | – | $3.25 \pm 0.68$ V | CAFM |
| | Supplier 6 | Specification | Not specified | – | Not specified | – | Unknown | – |
| | | Measured | 100% | CAFM, SEM | Not measured | – | $7.29 \pm 0.79$ V | CAFM |
| | Supplier 7 | Specification | Not specified | – | Not specified | – | Unknown | – |
| | | Measured | 100% | CAFM, SEM | Not measured | – | $5.74 \pm 0.80$ V | CAFM |
| | Supplier 9 | Specification | Not specified | – | ≥10 nm | Not specified | 10 V* | CAFM |
| | | Measured | 100% | CAFM, SEM | Not measured | – | $4.32 \pm 1.09$ V | CAFM |

The table summarizes the coverage and thickness of each sample as indicated by each supplier (in the product specifications) and measured in our lab. For the monolayer samples from Suppliers 1 and 2, we measured the physical thickness via cross-sectional transmission electron microscopy (TEM); as this technique is expensive and time-consuming, for the other monolayer samples we evaluated the thickness through the parameter onset potential ($V_{ON}$) and compared it with the $V_{ON}$ measured in mechanically exfoliated monolayer h-BN (see Fig. 1f)[25], highlighted in this table with the symbol "*". The value of $V_{ON}$ is defined as the minimum voltage detected when the current just exceeds the noise level (~3 pA). For the multilayer samples, Suppliers 2, 3 and 8 did not offer multilayers, so they are not in the table. Suppliers 5–7 did not specify the thickness, so we cannot compare $V_{ON}$. For the samples from Supplier 1, we measured the physical thickness via cross-sectional TEM; and for the samples from Suppliers 4 and 9, we compared the $V_{ON}$, as in the case of monolayer samples from Suppliers 3–9.

weak and inhomogeneous signal, with a wide FWHM of 16 cm$^{-1}$, plus at some locations (5 out of 12) no E$_{2g}$ band around 1367 cm$^{-1}$ is detected. These data demonstrate that the differences observed between the mechanically exfoliated monolayer h-BN sample (Fig. 1f) and the CVD h-BN from Supplier 1 (Fig. 3f) are related to the presence of atomic defects, pinholes, and thickness fluctuations in the CVD h-BN from Supplier 1.

We repeated the same experiments on CVD h-BN samples from Supplier 2 labelled as monolayers. Again, we collect 25 consecutive cross-sectional TEM images with atomic resolution. Our results indicate that 80.42% of the h-BN length analysed is, in fact, trilayer (i.e., 1.27-nm-thick) and that the other 19.58% shows the defective structure (see Fig. 3g, h and Supplementary Fig. 15). Large-area SEM images show long wrinkles (see Fig. 3i), which give the impression that the sample is continuous. The SEM images also show repetitive light-grey patterns all around the surface of the sample; these features are thicker and more insulating h-BN regions that cover 9.87% of the surface, often referred to as adlayers. When analysing the samples via CAFM, the surface roughness of the h-BN sample (provoked by the underlying Cu foil) is 8.90 nm, and the current maps collected without applying any

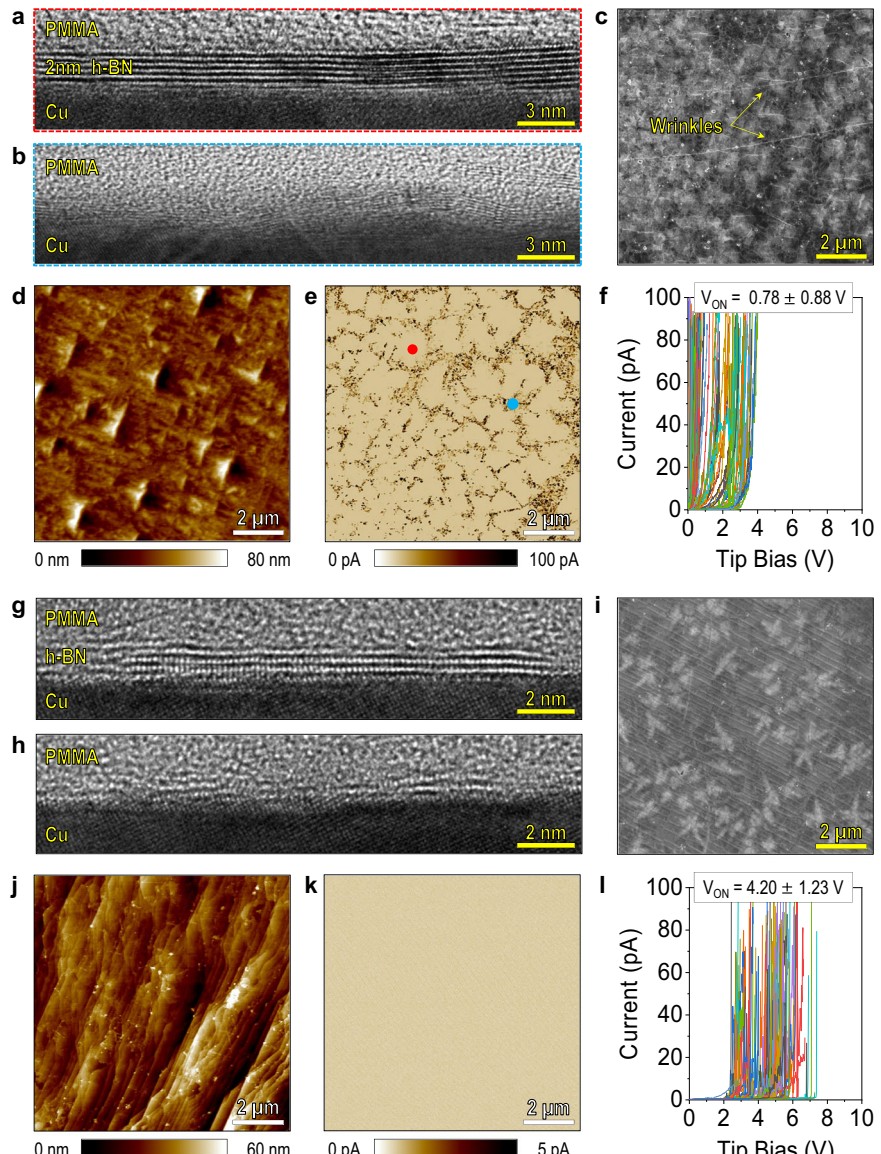

**Fig. 3 | Characterization of commercial CVD-grown h-BN samples labelled as monolayer from Suppliers 1 and 2.** Panels **a**–**f** and panels **g**–**l** are taken in samples from Suppliers 1 and 2 (respectively). **a**, **b** and **g**, **h** Cross-sectional high-resolution transmission electron microscopy (TEM) images of different areas of the CVD-grown h-BN samples labelled as monolayer from Suppliers 1 and 2. For both samples, ~80% of the sample looks like panels **a** and **g** (~2 nm-thick), and ~20% like panels **b** and **h** (amorphous); no monolayer area is detected. **c**, **i** Top-view scanning electron microscopy (SEM) images of the samples, showing long wrinkles and some brighter (i.e., multilayer) regions. **d**, **e** and **j**, **k** CAFM topography and current maps were collected simultaneously at the same area of each sample without applying any bias. **f**, **l** 100 $I$–$V$ curves collected at random locations of each sample.

bias show no current (Fig. 3k), meaning that the sample is free of pinholes. When applying RVS with the tip of the CAFM at 100 different locations of the sample, we observe that the h-BN is insulating at low voltages (see Fig. 3l), consistent with the current map in Fig. 3k. However, the value of $V_{ON}$ is very large and variable ($4.20 \pm 1.23$ V) indicating the presence of severe thickness fluctuations in the h-BN stack.

To have a more complete and global overview, we also characterized the CVD-grown samples labelled as monolayers from Suppliers 3–9 (see Supplementary Fig. 9) via SEM and CAFM. The data collected for each of them show significant variations of surface morphology, probably related to the different roughness of the Cu foils, generated provoked by the different recipes used to grow the h-BN, including pre-growth annealing. While this is an unwanted phenomenon, it is not against the specifications provided by the manufacturer, as the h-BN sheets could still be monolayer. Next, we did not

collect cross-sectional TEM images of the samples from Suppliers 3–9. The reason is that this type of experiment is especially expensive and time-consuming, not only due to the limited availability of high-resolution TEM equipment but also due to the lamella preparation process via FIB. Instead, we evaluated the thickness of the h-BN insulating films by collecting RVS via CAFM—a method proved to be valid to compare relative thickness variations[25]—and quantifying $V_{ON}$, which in theory (if the material is truly monolayer and free of defects) should be $0.21 \pm 0.08$ V (like in the case of mechanically exfoliated monolayer h-BN, see Fig. 1f). Our experiments indicate that CVD-grown h-BN labelled as monolayer by Suppliers 3–9 show values of $V_{ON}$ ranging from $1.51 \pm 1.02$ V (Supplier 9) to $6.94 \pm 1.33$ V (Supplier 6), as shown in Supplementary Fig. 9. This means that the samples are much more insulating, probably due thicker nature of the h-BN (as in the case of Suppliers 1 and 2), which ultimately does not meet the expected electrical properties of monolayer h-BN. The Raman spectroscopy

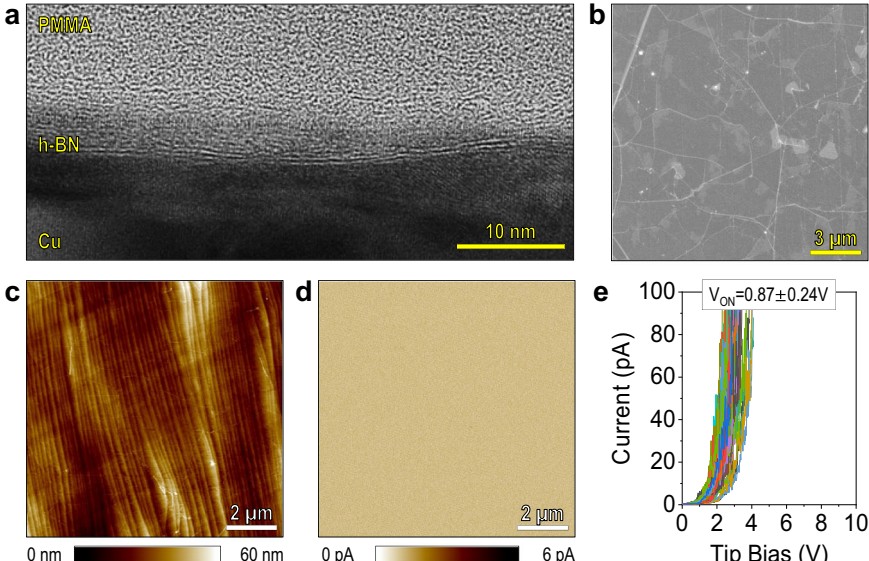

**Fig. 4 | Characterization of in-house CVD-grown bilayer h-BN grown in our lab. a** Cross-sectional high-resolution TEM image of the CVD-grown bilayer h-BN sample. **b** SEM image of the bilayer h-BN/Cu sample. **c** and **d** CAFM topography and current maps current without applying bias, respectively. **e** 100 *I–V* curves taken at random positions, collected by a CAFM; the currents have been limited to 100 pA to avoid degradation of the tip and the sample.

characterization has been repeated on all samples tagged as "monolayer" from Suppliers 2–9, by transferring the h-BN films onto 300 nm $SiO_2/Si$ substrates (see Supplementary Fig. 16). A clear and sharp $E_{2g}$ band of h-BN at $1367.02\,cm^{-1}$ can be observed for all the samples in their Raman spectra, with a wide FWHM various from 16 and $35\,cm^{-1}$.

## Homemade CVD-grown ultra-thin h-BN

We repeated the same experiments in an ultra-thin h-BN grown by the CVD method in our laboratory (see the "Methods" section). The sample is mainly bilayer (Fig. 4a), its surface is very clean (Fig. 4b, c), it is free of pinholes (Fig. 4d), and its $V_{ON}$ under RVS is $0.87 \pm 0.24$ V (Fig. 4e)– lower and more homogeneous than for all commercially available CVD-grown samples labelled as monolayer analysed in this study. Unfortunately, we cannot make a direct comparison between our homemade CVD-grown bilayer h-BN and mechanically exfoliated bilayer h-BN; the reason is that we did not succeed in fabricating mechanically exfoliated bilayer h-BN because this is an uncontrollable process, and we simply did not find any bilayer flake despite trying multiple times. Nevertheless, the value of $V_{ON}$ of our CVD-grown bilayer h-BN ($0.87 \pm 0.24$ V) is slightly higher and a bit more inhomogeneous than that of mechanically exfoliated monolayer h-BN ($0.21 \pm 0.08$ V), as it would be expected, and no large values and deviations (as in all commercially available samples) are detected. That demonstrates the high quality of our homemade CVD-grown bilayer h-BN.

## CVD-grown multilayer h-BN

Next, we analysed the quality of CVD-grown multilayer h-BN from Suppliers 1, 4–7 and 9 (the other suppliers did not offer multilayer CVD h-BN), discussing mainly surface roughness, average thickness, thickness fluctuations, number of atomic defects, observation of pinholes and electrical homogeneity (through $V_{ON}$). For Supplier 1, we characterize 5 different samples so that we can also analyse sample-to-sample variability. We selected Supplier 1 for the variability analysis because its samples are the most used in the literature (see Fig. 2). The samples from Supplier 1 were purchased on the same day and arrived at our laboratory, although they were grown in different batches, as confirmed by the manufacturer.

Under the optical microscope, all the samples look continuous, and no optical contrast is observed (see Supplementary Fig. 23), indicating that the thicknesses of all the CVD-grown multilayer h-BN samples (whatever it is) seem to be homogeneous throughout each sample. Figure 5 shows the SEM and CAFM data collected for the samples from Suppliers 4–7 and 9, and Fig. 6 shows the TEM and CAFM data collected for the 5 samples from Supplier 1 (additional SEM and TEM images of the samples from Supplier 1 is shown in Supplementary Figs. 17–22). SEM and CAFM inspections reveal that the surface roughness of all multilayer CVD h-BN/Cu samples is higher than for the "monolayer" ones. We cannot be sure if the suppliers intentionally treated the surface of the Cu foil to increase its roughness and prevent the formation of wrinkles[26] or if this is related to the larger growth time. From an electrical point of view, none of these multilayer samples show pinholes, as current maps collected when applying 0 V show no current spots (see an example in Supplementary Fig. 24).

Next, we statistically evaluate $V_{ON}$ for all the commercially available CVD-grown h-BN samples labelled as multilayers by collecting 100 *I–V*s at different locations. Supplier 4 indicated in its specifications that the thickness of the multilayer h-BN is 13 nm (see Table 1), which (if the sample would be of high quality) should result in $V_{ON}$ well above 12 V because 5.65-nm-thick h-BN exhibited $V_{ON} = 7.85 \pm 0.50$ V (see Supplementary Fig. 5f). However, the sample from Supplier 4 exhibits $V_{ON} = 5.34 \pm 0.47$ V, which is much lower and indicates that the sample is either much thinner or contains many more local defects that increase the leakage current by trap-assisted tunnelling. Similarly, the sample from Supplier 9 (labelled as multilayer h-BN with a thickness >10 nm) shows $V_{ON} = 4.32 \pm 1.09$ V, which is much lower than the expected values (around 10 V, see Supplementary Fig. 5f). These observations indicate that the CVD-grown multilayer h-BN samples from Suppliers 4 and 9 are much more conductive than mechanically exfoliated h-BN samples of similar thickness. We cannot know if this discrepancy would be considered acceptable by the supplier, as no acceptable variability ranges of thickness are indicated in the specifications. In the case of Suppliers 5–7, the vendors did not specify the expected film thickness, and hence we cannot know if the values of $V_{ON}$ measured ($3.25 \pm 0.68$, $7.29 \pm 0.79$, and $5.74 \pm 0.80$ V) are suitable.

When measuring current maps at 0.5 V, the samples from Suppliers 5 and 9 show some weak spots (the leakage current is higher at

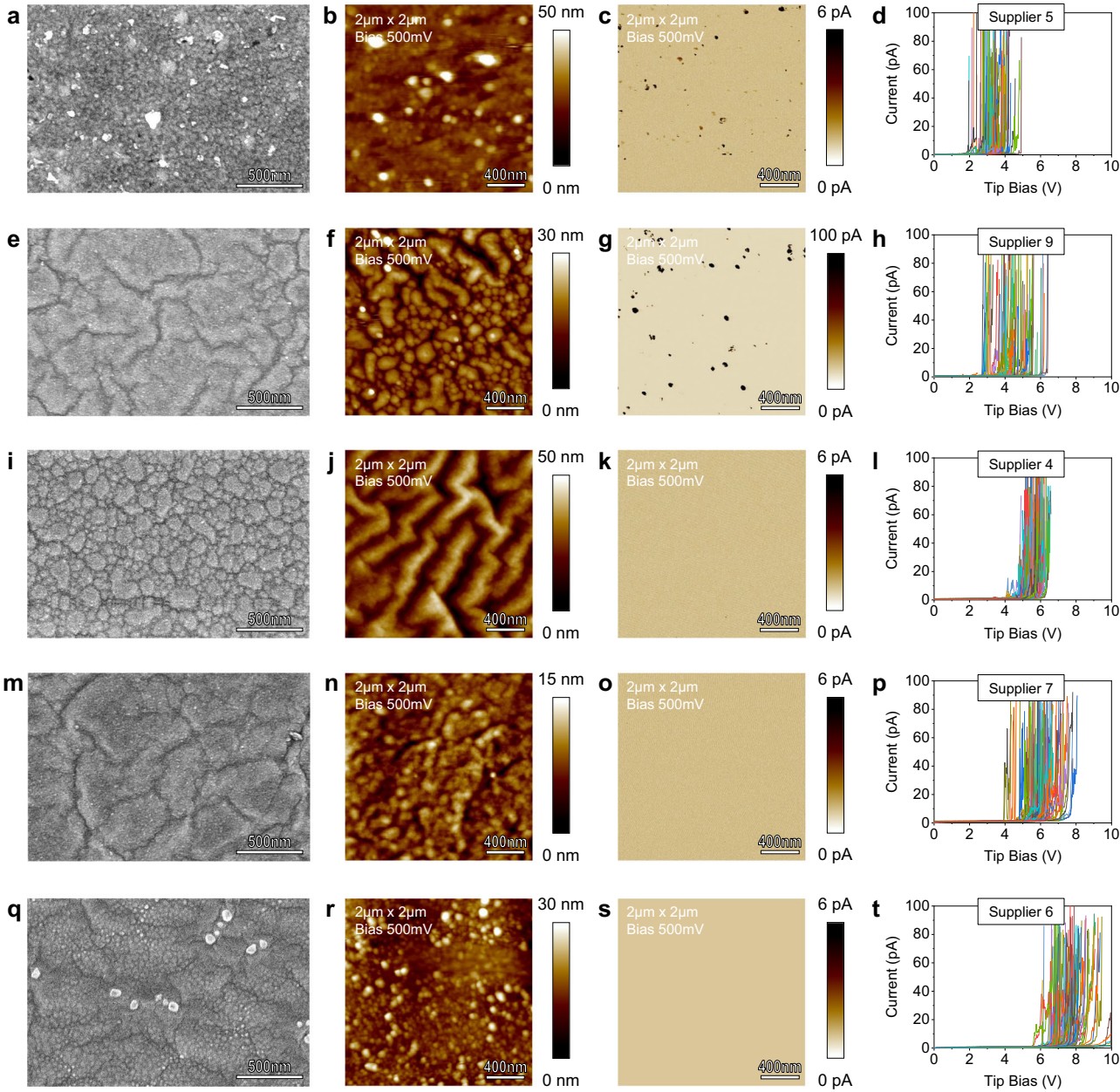

**Fig. 5 | Characterization of commercial CVD-grown multilayer h-BN from Suppliers 4–7, and 9.** The results from different suppliers are ordered from the most conductive to the most insulating (detected by the $V_{ON}$). The first column (**a**, **e**, **i**, **m**, and **q**) shows the SEM images of each sample, the second (**b**, **f**, **j**, **n**, and **r**) and third (**c**, **g**, **k**, **o** and **s**) columns show the CAFM topography and current maps at 0.5 V, respectively, and the right column (**d**, **h**, **l**, **p**, and **t**) shows 100 forward I–V curves for each sample collected at random positions (in a matrix of 10 μm × 10 μm).

those locations), which can represent a problem if the material is used as gate dielectric in transistors, but it has been shown to be beneficial to fabricate memristors. The Raman spectra of multilayer CVD-grown h-BN from Suppliers 4–7 and 9 (transferred onto 300 nm $SiO_2$/Si substrates) show $E_{2g}$ band of h-BN at 1367.02 cm$^{-1}$ for all the samples, but the signals are especially weak in multilayer h-BN samples from Suppliers 4 and 9, with a wide FWHM of 36 and 38 cm$^{-1}$, respectively (see Supplementary Fig. 25). The inhomogeneous Raman signal shown in the Raman mappings indicate their thickness fluctuation and high density of local defects.

Next, we analyse the variability of morphological and electrical properties in five different CVD-grown multilayer h-BN samples from Supplier 1. We collected 25 consecutive cross-sectional TEM images for each sample from Supplier 1 and all (100%) of the images show traces of layered structure, although they host large amounts of local

defects and lattice distortions (see Fig. 6 left column and Supplementary Figs. 18–22). The layered structure seems to be more accentuated in the thinnest sample (Fig. 6a), and for the others, the interfaces with C and Cu are blurry, the layers seem to be interrupted and have bifurcated, and for the thickest samples even oblique and almost vertical layers can be observed. This behaviour has also been observed in 2D materials grown by other academics[27]. A bigger problem is that the thickness of the h-BN is different for each sample, ranging between 3 and 5 layers for the thinnest (~1.5 nm, Supplementary Fig. 19) and between 18 and 22 for the thickest (~6 nm, Supplementary Fig. 22). None of the samples offers a thickness that approaches to the 13 nm advertised on the website of Supplier 1 for this type of samples. The density of native defects is also different for each sample, and it seems to increase with the thickness (i.e., in thicker samples, the top layers are more defective and even show

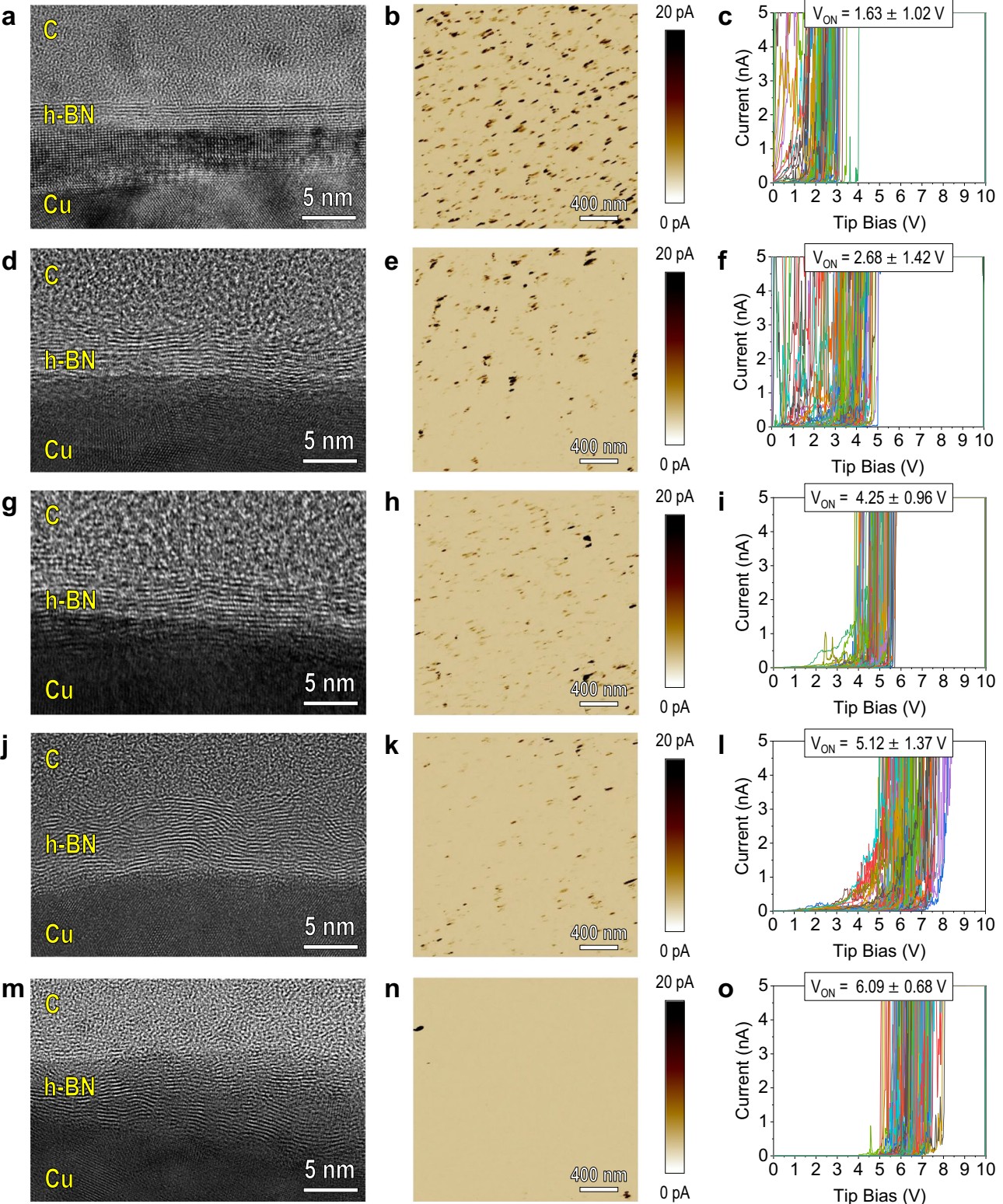

**Fig. 6 | Characterization of commercial CVD-grown multilayer h-BN from Supplier 1.** Left column (**a**, **d**, **g**, **j**, and **m**) shows the cross-sectional TEM images of five different samples. The central column (**b**, **e**, **h**, **k**, and **n**) shows the CAFM current maps at 0.5 V, and the right column (**c**, **f**, **i**, **l** and **o**) shows 100 *I–V* curves for each sample collected at different positions of the sample (in a matrix of 10 μm × 10 μm).

inclined/vertical alignment, as it happens in HfO$_2$ and other high-*k* dielectrics[28]).

The CAFM current maps collected without voltage show no current at all, indicating that all the samples are free of pinholes (like Supplementary Fig. 24). Current maps collected under 0.5 V for all the samples show that the thinnest h-BN drives substantial current (above 1 nA, see Fig. 6b), while the thickest one drives no current (see Fig. 6n). When applying RVS we observe values of $V_{ON}$ ranging from 1.63 ± 1.02 V for the thinnest sample to 6.09 ± 0.68 V for the thickest (see Fig. 6, right column). Note that CVD h-BN stacks were

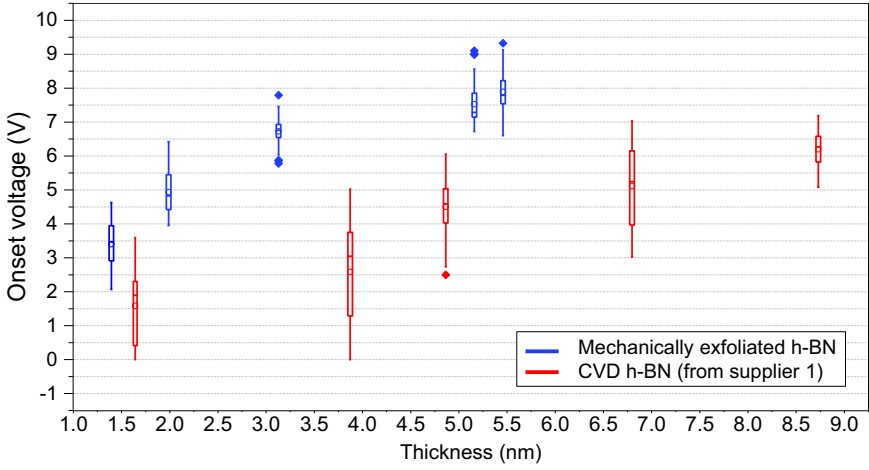

**Fig. 7 | Benchmarking the insulating properties of CVD h-BN.** Statistical analysis of the onset potential registered for 100 $I$–$V$ curves measured at different random locations of mechanically exfoliated h-BN and CVD-grown h-BN (from Supplier 1) for different thicknesses. Each box includes 100 data points. Inside each box, a thick solid line indicates the median value of the $V_{ON}$ calculated from the 100 $I$–$V$ curves, while the error bar represents the standard deviation of the $V_{ON}$ calculated from the same 100 data points.

characterized on their Cu substrate employed for the growth (no polymer residues on the surface of the samples and no gaps between the h-BN and the substrate), implying that the large differences observed should be related to the batch-to-batch variability of thickness and native defects. It is worth noting that there is a very good correlation between thickness (observed in TEM images) and the value of $V_{ON}$ (observed in $I$–$V$ plots), which indicates that h-BN thickness plays the most important role (even over crystallinity and density of bonding defects).

Combining the results in mechanically exfoliated h-BN samples with different thicknesses (see Supplementary Fig. 5), Figure 7 shows a general overview of $V_{ON}$ for all the mechanically exfoliated h-BN samples and all the CVD-grown h-BN samples (from Supplier 1), versus their real thickness (measured via cross-sectional TEM images). The data indicates that commercial CVD h-BN samples from Supplier 1 are more conductive and inhomogeneous (lower mean and higher deviation of $V_{ON}$) than the mechanically exfoliated counterparts for all thicknesses analysed above ~1.3 nm.

## Discussion

We analyse commercially available CVD-grown h-BN samples from nine different suppliers—all the vendors available on the market at the time of the study—and found that the morphology and structural properties of their samples do not match the product specifications. The samples labelled as "monolayer" by Suppliers 1 and 2 are found to be thicker (~2 nm thick and ~1.27 nm thick, respectively). Moreover, in the case of Supplier 1, ~20% of their volume is not layered but amorphous (which sometimes produces short circuits), and they contain multiple thicker islands (i.e., adlayers). The thickness of several samples offered as multilayer (from 2 to 7 nm) differs a lot from the typical values specified on the website of Suppliers 1, 4 and 9 (13 nm in average or ≥10 nm). Moreover, the density of native defects in CVD h-BN is very large, which results in much higher leakage current than in mechanically exfoliated h-BN samples of similar thicknesses. Several suppliers of CVD-grown h-BN label the samples as "multilayer" without indicating any specific thickness, which does not allow its use to fabricate devices with controllable properties. We hope that our study helps researchers to better understand the quality of commercially available CVD h-BN and that it serves as a call to action for companies to (in the short term) provide more accurate product specifications on their websites and (in the middle-long term) improve the quality of the samples offered.

## Final note

Suppliers 1–4 and 6–9 did not provide any statement when asked to comment on the findings reported in this Article.

Supplier 5 provided the following statement:

"A key point to note is the substantial variability in h-BN growth across different batches. This variance has been consistently observed by our company and by clients from various organizations in the past. Consequently, any direct comparison of h-BN sourced from different vendors lacks completeness and fails to capture the broader context. This stands as a primary concern regarding your manuscript.

Additionally, we would like to address the temporal behaviour of our products and those of other vendors. In addition to the batch-to-batch differences, our company has noted a time-dependent response in the performance of manufactured 2D layers, including h-BN. Specifically, within a given batch and company, samples tend to degrade over time. In the case of h-BN, this degradation is attributed to oxidation at the h-BN/Cu interface and oxygen diffusion at the grain boundaries."

## Methods

### Materials

The CVD h-BN samples were grown on Cu foils, and we characterized them without transfer. We prefer not to disclose the names of suppliers of CVD h-BN employed in this study, although more information about the usage of those samples is presented in the caption of Fig. 2. The reference h-BN sample produced by mechanical exfoliation has been prepared using small h-BN crystals from the National Institute for Materials Science in Japan. The mechanically exfoliated samples were transferred, using a transfer stage from HQ graphene, on a 5 nm Ru/30 nm Ta/300 nm SiO₂/Si substrate. The CVD h-BN samples grown in our laboratory use standard ~25-μm-thick Cu foils from Thermo Fisher Scientific as substrate. After being introduced in the tube, the temperature is raised to 1050 °C for 30 min in an H₂ (10–20 sccm) and Ar (5–10 sccm) atmosphere for annealing and surface cleaning. After that, the valve that controls the flow of the precursor (ammonia borane, $H_3NBH_3$, 95%, from Sigma-Aldrich) into the chamber is opened, and the temperature is adjusted at 80 °C during the h-BN growth. Finally, the precursor valve is closed, and the temperature is ramped down to room temperature naturally. The SiO₂ samples were synthesized via thermal oxidation on n⁺⁺ Si substrates in an industrial foundry.

## Characterization

The SEM machine employed is a Helios G4 UX, the TEM is an FEI Titan Themis[3] S/TEM, and the FIB used to prepare the samples is a Helios G4 UX. The electrical measurements were carried out using a Dimension Icon atomic force microscope from Bruker in contact mode under an ambient atmosphere. The sample was placed on a metallic plate for CAFM measurements, and the Ru film was connected to the plate using conductive Ag paint. All the measurements were performed with Pt-coated Si tips from Bruker (CONTV-PT, nominal tip radius below 25 nm, spring constant -0.2 N/m, length of cantilever 450 μm). The defection setpoint used in the experiments are: +1 V during engagement, 0 V during scanning. The scan rate is 1 Hz (while scanning areas of 2 μm × 2 μm, 5 μm × 5 μm and 10 μm × 10 μm). The current in each RVS was limited to 100 pA, which prevents tip degradation and ensures the reliability of the data collected. We checked the repeatability of our experiments by comparing the tip's conductivity before and after the measurements and no difference could be observed; hence, tip degradation can therefore be excluded. Raman measurement was performed by confocal Raman spectroscopy (Alpha 300R, WITec) with a laser wavelength of 532 nm.

We would like to further emphasize that the use of CAFM in contact mode (with the types of tips and contact forces employed in this study) does not damage the surface of the CVD-grown h-BN samples. To confirm this, we make a test experiment that consists of collecting one 15 μm × 15 μm topographic scan in contact mode at a random location of a h-BN sample transferred on a SiO₂/Si substrate. Then, we zoom-in and scan 15 times with a size of 10 μm × 10 μm, and finally, we zoom-out and scan again with a size of 15 μm × 15 μm. Despite the wrinkles in the 2D material are known to be soft regions that could be displaced if enough force is applied[29], all the images collected (see Supplementary Fig. 26) are identical and clearly demonstrate that no damage or surface modification is introduced by the tip of the CAFM.

## Data availability

Relevant data supporting the key findings of this study are available within the article and the Supplementary Information file. All raw data generated during the current study are available from the corresponding authors upon request.

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

## Acknowledgements

This work has been supported by the generous Baseline funding scheme of the King Abdullah University of Science and Technology. Prof. Mario Lanza acknowledges the platform Web Of Talents (https://weboftalents.com) for support in the recruitment of talented students and postdocs.

## Author contributions

M.L. conceived the idea, designed the experiments and supervised the entire investigation. Y.Y. fabricated the exfoliated h-BN samples and characterized all of the h-BN samples. J.W. characterized the SiO₂ samples and helped with the data statistical analysis tasks. J.L., B.T. and X.Z. fabricated the in-house CVD-grown ultra-thin h-BN. Y.M. fabricated the flat metal substrates for mechanically exfoliated samples. Y.Y. and Y.M. performed the Raman experiments. T.T. and K.W. provided the h-BN crystals, which were used for the exfoliation. M.L. and Y.Y. wrote the manuscript. All the authors read the manuscript and provided comments.

## Competing interests

The authors declare no competing interests.
