## [Peer Review File · Nature Communications]

On the quality of commercial chemical vapour deposited hexagonal boron nitrideEditorial Note: Parts of this Peer Review File have been redacted as indicated to a). preserve confidential or personal information and b). remove third-party material where no permission to publish could be obtained.

REVIEWER COMMENTS

Reviewer #1 (Remarks to the Author):

In this paper, the authors used the cross-sectional TEM and C-AFM techniques to evaluate the thickness and crystal/electrical quality of the commercially-available “monolayer” and “multilayer” CVD h-BN films. With great efforts, they claimed that some of the most popular suppliers of CVD h-BN provide samples which properties are very far from those advertised in their product specifications. Although this paper can be regarded as the excellent guidance of quantitative electrical analysis on ultrathin insulators by C-AFM, the referee thinks the significance and impact of this work still can not meet the high standard of Nature Communications. The main reasons are listed as follows:

1) It is a widely believed fact among 2D community that the quality of CVD-grown 2D materials is typically lower than the mechanically exfoliated one. Therefore, the referee is not surprising to the authors' claim. This paper seems like an accuse of false propaganda on CVD h-BN products of some suppliers, rather than claims all commercial CVD h-BN is of poor quality. However, the names of suppliers of CVD h-BN employed in this study are not disclosed.

2) This paper mainly focuses on the breakdown strength of CVD h-BN via C-AFM, which is a key parameter for the dielectric layer. However, at current stage, it is the mechanically exfoliated h-BN, rather than the CVD h-BN, is regarded as vdWs dielectric layer to fabricate hysteresis-free and ultrahigh-mobility heterostructure electronic devices. Obviously, the CVD h-BN grown on rough Cu foil is not compatible to the ideal and clean vdW integration with other 2D materials, since contaminations are inevitable during the transfer process. To this end, the referee thinks no one worries about the pinholes in commercially CVD h-BN, but the pinholes of mechanically exfoliated h-BN matters. Hence, if the authors tell us that the commercially-available h-BN bulk crystals are all far from satisfactory to fabricate high-performance h-BN based devices, it warrants publication in Nature Communications.

The following remarks and questions may be helpful to the authors.

1) Typically, the breakdown strength of a dielectric insulator is proportional to its band gap. h-BN has a wide band gap of ~ 6 eV, which should have a large breakdown voltage even for the ultrathin samples. However, as shown in Fig. 2f, all the mechanically exfoliated h-BN with thickness ranging from monolayer to tri-layer are not insulating at all. Why? It seems like the quality of the mechanically exfoliated h-BN is poor. The authors should do extra experiments, such as fabricating MIM capacitors, to figure out the quality of mechanically exfoliated samples, and compare the breakdown voltage of MIM capacitors to the one measured by C-AFM.

2) The C-AFM is a very local technique for I-V measurements based on contact mode, in which surface damages are inevitably caused during the tip scanning. Therefore, it may be not a suitable tool for quantitatively determining the amounts of pinholes and defects of ultrathin dielectric layers. To remove the possible influence, laminating another layer of thin Au nanoflakes on top of h-BN is suggested. In this case, the AFM tip will not directly contact the thin dielectric layers.

3) The authors seemed to believe that lower V_{on} means higher quality of h-BN. Theoretically speaking, perfect crystals should have higher breakdown voltage. The CVD grown h-BN has a much higher V_{on} .

Does that mean the quality of monolayer CVD h-BN in author's lab is higher than the mechanically exfoliated one?

4) Surface roughness on Cu foil is much higher than the one on Ru film. Does it affect the Von measured?

For a rough surface, the contact area between the surface and tip are relatively small.

5) In Fig. 3e, the authors attributed the conducting area to the amorphous h-BN. However, since no in-situ TEM data were performed, the authors should adopt a more conservative saying.

6) In Fig. S1c, please explain the steps and noise burr observed. Besides, please use the standard resistance (for example 1 M Ohms) to determine the offset voltage and noise of narrow voltage scan.

Reviewer #2 (Remarks to the Author):

This paper reports the benchmarking of commercially available CVD-grown hBN (both monolayer and multilayer), comparing with monolayer hBN grown in the authors' lab and multilayer hBN exfoliated from NIMS hBN. The quality assessment of commercial hBN samples is very important for many researchers, because hBN is a key insulating material for 2D research. The authors found that all the commercially available CVD-hBN products do not meet the specifications supplied from the companies.

I think that most of the readers are interested in which company sells the best quality hBN or in the actually quality, such as thickness, crystallinity, of the hBN samples which they are using or they plan to use. However, in this manuscript, the authors do not provide the company names (though Fig. 1 is somehow related to the company name), thus the readers cannot get enough information related to their research. Furthermore, since the authors simply examined cross-section TEM images, CAFM data, and SEM images of different samples, it lacks scientific discussion how the hBN quality comes from each CVD process. In other words, scientific insight and findings are not new enough. Therefore, I think that this manuscript does not meet the high standard of Nature Communications and, thus, cannot recommend the publication.

The followings are some small comments on this manuscript.

1. The details of the CVD growth of monolayer hBN in the authors' lab is not well described in the experimental section. Please describe the details of the CVD synthesis, such as the feedstock, CVD temperature, and Cu foil source, because their hBN is used as the standard monolayer sample.

2. The TEM image of their monolayer hBN (Fig. 4a,b) looks like bilayer hBN. Is this really monolayer? More details explanation is required.

3. Figure caption of Fig. 6 is incorrect. "suppliers from 3-9" (from 7 companies) should be "suppliers from 4, 5, 6, 7, 9" if I can understand the explanation in the main text.

4. It is better to describe why they used Ru/Ta as an electrode.

Reviewer #3 (Remarks to the Author):

The manuscript by Yuan et al. reports the structural and electrical characterization of multilayer and “monolayer” CVD-grown h-BN film samples supplied by the main commercial suppliers, and the results are compared with mechanically exfoliated flakes based on h-BN crystals provided by the NIMS group as a reference of high-quality (standard) h-BN films. The statistical analysis presented here is of great value for the community and the conclusions of the work are clear: the properties of CVD h-BN films provided by the most popular suppliers are very far from those advertised in their product specifications, regarding film thickness, crystal quality, homogeneity and most critically, electrical properties. In general, I think the conclusions reached by Yuan et al. may serve as a starting point to bring common understanding on the actual quality of samples provided by commercial suppliers, and hopefully to speed up the process of standardization in the market of commercial 2D-materials. However, whether this study deserves publication on Nature Communications or other more specialized journals, I would leave this to the editor to judge, as I do not find any new science in this paper. Following are some comments the authors should address before publication, either on Nature Communications or elsewhere.

1. The main use of h-BN in 2D research is as a dielectric layer, and it has been shown that h-BN encapsulation can drastically increase the mobility of 2D devices. I think it would be more useful to construct devices using various h-BN samples and compare the device performance.
2. The cross-sectional TEM analysis, while is quite useful for thickness measurement, is not typically considered an efficient method for statistical analysis, as the experiment is time consuming and expensive while each sample can only cover regions at a scale of 5-20 microns. It would be more useful if the authors can perform other statistically more efficient analysis, e.g. Raman mapping, to characterize the homogeneity of the samples.
3. In the analysis of cross-sectional TEM (for example Figure 3a-b), how can we be sure the “amorphization” is not induced by the FIB sample preparation? For example, in Figure 3b, the Cu foil also looks remarkably damaged as compared to Figure 3a. How to make sure this is the original state of the films after sample preparation? The FIB process can easily damage ultrathin films, especially if they are composed of light elements such as C or B-N. In addition, some of the samples seem to be protected only by amorphous carbon layers. It should be pointed out that even e-beam deposited carbon or thermal evaporated carbon can damage the surface structure of 2D materials. The authors need to be very careful during their sample preparation.
4. More on this, the authors mention several times throughout the manuscript the possible presence of defects, especially in the “amorphous” areas, but no clear analysis is provided. Do the authors imply here that defective areas are prone to amorphization and is the cause of the observed “amorphous” areas? Seems difficult to correlate both ideas without proper structural analysis. In addition, from the cross-sectional TEM results, one can only describe e.g. the region in Figure 5j as distorted or disordered regions, because such images do not provide conclusive information about the in-plane crystallinity. So, calling it as “amorphous” may not be correct.
5. In Figure 2f, the authors mention that the I-V measurements in mechanically exfoliated monolayer h-BN show good agreement with the calculated curves for monolayer and bilayer h-BN. However, I see that the vast majority of the grey curves (experiments) correlate with the blue line, which corresponds to a trilayer model. Why is this discrepancy in the interpretation? Furthermore, in the following section (3. CVD-grown “monolayer” h-BN), the monolayer model is assumed as reference according to the

abovementioned interpretation in Figure 2f. Is this whole interpretation correct and how is this affecting the results from commercial samples?

6. The authors did not discuss the huge increase in VON for “monolayer” CVD h-BN samples from supplier 2. What is the cause?

7. Figure 4 shows the characterization of in-house CVD-grown monolayer h-BN. However, from the TEM image I can count at least 2 layers, and seemingly up to 4 layers at some points. Despite the films seem to have better crystalline quality than those provided by commercial suppliers, the statement of “truly monolayer” is incorrect. Why in this case the VON is the lowest even if not a monolayer?

8. The manuscript seems to be written in a rush and lacks many important experimental details. The figure captions were not carefully written, without clear explanation of each panel, and one needs to refer to the main text in order to understand the details. Figure 6 says results from suppliers 3-9, but one has to find out from the fine print in the last column that these samples are from suppliers 5,9,4,7,6, in a very odd arrangement. This is particularly annoying.

9. Section 4 needs to be rewritten and put the information in a more coherent way. It is quite hard to read and follow. For instance, in the beginning the authors mention they want to study the variability from the multilayer samples provided by supplier 1, but one cannot know which parameter are they studying. Looking at Figure 5, one can guess is thickness, but it is confusing since those are supposed to be multilayer samples with thickness >10 nm. The text and the description of the results is incomplete and not possible to follow clearly till the end of the section, where the text reveals there is a thickness mismatch between samples. The authors should rewrite this section of the manuscript. In the same line, it is confusing that the authors describe Figure 6 before figure 5.

10. In the second paragraph of Section 4, CAFM images of samples from supplier 5 and 9 show pinholes, but in line 3, the authors claim samples from supplier 9 was pinhole free. In addition, in paragraph 3 the authors say Samples from supplier 1 are free of pinholes while Figure 5b-n clearly show the same patterns as in Figure 6. How to understand this?

11. If the authors try to alert the community about the real quality of the commercially available samples, details about the suppliers should be provided. Otherwise, it hurts all suppliers.

Answers to the comments from reviewer #1

In this paper, the authors used the cross-sectional TEM and C-AFM techniques to evaluate the thickness and crystal/electrical quality of the commercially available “monolayer” and “multilayer” CVD h-BN films. With great efforts, they claimed that some of the most popular suppliers of CVD h-BN provide samples which properties are very far from those advertised in their product specifications. Although this paper can be regarded as the excellent guidance of quantitative electrical analysis on ultrathin insulators by C-AFM, the referee thinks the significance and impact of this work still cannot meet the high standard of Nature Communications.

We thank a lot to the reviewer for indicating that our manuscript can be regarded as an excellent guidance of quantitative electrical analysis on ultrathin insulators by C-AFM.

The number of publications on CVD-grown h-BN and the citations to those publications clearly indicate that this topic is very significant for a wide community. [REDACTED]. Regarding impact, the quality of the CVD-grown h-BN in terms of thickness fluctuations, number of lattice defects, tunnelling current and electrical homogeneity has an extremely high impact on the properties of electronic devices. Understanding the quality of commercial CVD-grown h-BN can be invaluable for a wide community of scientists employing this material. As our manuscript provides unique information in this direction, we believe it is both significant and impactful.

In the following we provide answers (including additional data when needed) to the technical comments raised by the reviewer. We think the comments raised by the reviewer have been useful and have helped to enhance the quality of our article, and hence we would like again to thank the reviewer for his/her careful revision and constructive feedback.

The main reasons are listed as follows:

1) It is a widely believed fact among 2D community that the quality of CVD-grown 2D materials is typically lower than the mechanically exfoliated one. Therefore, the referee is not surprising to the authors' claim. This paper seems like an accuse of false propaganda on CVD h-BN products of some suppliers, rather than claims all commercial CVD h-BN is of poor quality. However, the names of suppliers of CVD h-BN employed in this study are not disclosed.

We thank the reviewer for this comment. We knew before starting to write this manuscript that CVD-grown h-BN possesses lower quality than mechanically exfoliated. This is known by the community, as the reviewer well pointed. This kind of thinking has been gained from device-level studies. For example, when using mechanical exfoliation, graphene transistors with h-BN substrate show mobilities of $60,000 \text{ cm}^2\text{V}^{-1}\text{s}^{-1}$ (see Dean et al. Nature Nanotechnology 5, 722-736 2010), and when the same experiments are repeated using CVD h-BN the mobility observed is only $2,500 \text{ cm}^2\text{V}^{-1}\text{s}^{-1}$ (see Pandey et al. IEEE Trans. Electron Dev. 65, 4129-4134, 2018). However, the quality of CVD-grown 2D materials evolves, and an updated benchmarking study with CVD h-BN samples from different commercial sources (which has never been done before) is necessary.

[REDACTED].

[REDACTED]

Figure R1 | Example of another study that reports on the quality of commercial 2D materials (in this case Graphene) without revealing the identity of each company. a, Graphene content per number of companies. **b,** Number of companies related with the number of layers from AFM (D50 and D90). Reproduced from Kauling, A. P. et al. *Advanced Materials* 30, 1803784, 2018.

2) This paper mainly focuses on the breakdown strength of CVD h-BN via C-AFM, which is a key parameter for the dielectric layer.

Our manuscript focuses on describing the electrical homogeneity of the samples, which is characterized by the onset potential (V_{ON}) measured during ramped voltage stresses and the density and size of conductive spots in lateral scans.

We have noticed that our manuscript had a sentence in the introduction saying: *“The main conclusion of our study is that the leakage current, electrical homogeneity and dielectric strength of commercially available CVD h-BN samples are significantly worse than those of mechanically exfoliated h-BN.”* That is a mistake from our side, we apologize for the confusion created.

In this revision, this sentence has been modified as follows: *“The main conclusion of our study is that the leakage current and electrical homogeneity of commercially available CVD h-BN samples are significantly worse than those of mechanically exfoliated h-BN.”* In other words, we have removed the concept breakdown strength, which now is not mentioned in the manuscript (as we didn't study such thing).

However, at current stage, it is the mechanically exfoliated h-BN, rather than the CVD h-BN, is regarded as vdWs dielectric layer to fabricate hysteresis-free and ultrahigh-mobility heterostructure electronic devices. Obviously, the CVD h-BN grown on rough Cu foil is not compatible to the ideal and clean vdW integration with other 2D materials, since contaminations are inevitable during the transfer process. To this end, the referee thinks no one worries about the pinholes in commercially CVD h-BN, but the pinholes of mechanically exfoliated h-BN matters. Hence, if the authors tell us that the commercially available h-BN bulk crystals are all far from satisfactory to fabricate high-performance h-BN based devices, it warrants publication in *Nature Communications*.

We disagree with this statement. The semiconductors industry only cares about the quality of CVD-grown 2D materials, as stated in multiple technology roadmaps (see *The International Roadmap for Devices and Systems*, 2021 Edition; <https://irds.ieee.org/editions/2021>). The quality of the h-BN film after the transfer is critical and it is exhaustively studied by many companies and centres, such as Imec (see S. Brems et al. in *Proc. 2023 Symposium on VLSI-TSA*, Hsinchu, Taiwan). There is no company using mechanical exfoliation for device fabrication, and hence, whether its quality is good or not has little impact on the semiconductors industry (although studies on mechanical exfoliation are interesting to know the limits of the material and used them for comparison purposes, as we did in our article).

The following remarks and questions may be helpful to the authors.

1) Typically, the breakdown strength of a dielectric insulator is proportional to its band gap. h-BN has a wide band gap of ~6 eV, which should have a large breakdown voltage even for the ultrathin samples. However, as shown in Fig. 2f, all the mechanically exfoliated h-BN with thickness ranging from monolayer to tri-layer are not insulating at all. Why? It seems like the quality of the mechanically exfoliated h-BN is poor. The authors should do extra experiments, such as fabricating MIM capacitors, to figure out the quality of mechanically exfoliated samples, and compare the breakdown voltage of MIM capacitors to the one measured by C-AFM.

The h-BN films show high leakage currents (despite having a large band gap of ~6 eV) because they are very thin: 0.33 nm for monolayer and 0.99 nm for tri-layer. Due to quantum tunnelling, currents can be measured when the thickness is lower than 1 nm, no matter how large the bandgap is (see Chu et al. *Advances in Materials Science and Engineering*, 2014, 578168, <https://doi.org/10.1155/2014/578168>). These currents measured have been observed by other authors (see Lee, G. et al., *Applied Physics Letters* 99, 243114, 2011 and Britnell, L. et al. *Nano Letters* 12, 1707-1710, 2012).

As mentioned, we are not studying dielectric strength in this manuscript. Instead, we are measuring onset potential, which is always characterized using CAFM. Hence, we don't see the need of measuring breakdown voltages in MIM capacitors in this study. Moreover, both the onset potential and the breakdown voltage measured via CAFM should never be compared directly with those measured in MIM capacitors, due to the different background currents related to the lateral size of the devices. Moreover, in CAFM the tip-sample contact force plays a very important role. This is explicitly mentioned in different CAFM books (see Celano et al. *Electrical Atomic Force Microscopy for Nanoelectronics*, Springer-Nature, ISBN 978-3-030-15611-4, 2019 and Lanza et al. *Conductive Atomic Force Microscopy: Applications in Nanomaterials*, 2017. Wiley-VCH, ISBN: 978-3-527-34091-0). Furthermore, the fact that one device contains more defects does not mean that the h-BN contains more native defects, as many defects can be introduced during device fabrication steps, such as metal evaporation on the 2D material (see for example Zheng et al. *Advanced Materials* 2022, 34, 2104138).

2) The C-AFM is a very local technique for I-V measurements based on contact mode, in which surface damages are inevitably caused during the tip scanning. Therefore, it may be not a suitable tool for quantitatively determining the amounts of pinholes and defects of ultrathin dielectric layers. To remove the possible influence, laminating another layer of thin Au nanoflakes on top of h-BN is suggested. In this case, the AFM tip will not directly contact the thin dielectric layers.

We disagree with this comment. Local techniques have been used for the detection of pinholes in 2D materials, as the group from Prof. Lain-Jong Li demonstrated using in scanning tunnelling microscopy (see Wan et al. *Nature Communications*, 13, 4149, 2022). Moreover, hundreds of reputed scientists all around the world use CAFM for the study of nanomaterials, and it has been even highlighted by the International Roadmap of Devices and Systems in its Metrology section (see *The International Roadmap for Devices and Systems. 2021 Edition* (accessed 04 February 2023); <https://irds.ieee.org/editions/2021>). CAFM has been proven to be a reliable technique that does not damage the surface of the sample if the experiments are done carefully.

In our study we have used a conductive tip with a low spring constant of 0.2 N/m. When we measure the same area of a sample multiple times with this type of tips, we observe zero damage, as seen by the presence of wrinkles in identical positions of the maps when measuring sequentially (see Figure R2a,d and R2b,e), and zoom-out maps also prove that there is no morphology modification in the sample surface (see Figure R2c,f). Lamination of the 2D material is only produced when using very stiff tips (such as diamond tips) and when using very high contact forces.

For clarity, in this revision we have repeated the experiments by applying such a high contact force with a solid diamond tip (model AD-40-AS tip, spring constant 40 N/m), and then we can see such lamination, as shown in Figure R3. Note that all the I-V curves in the CVD h-BN samples presented in our manuscript have been collected without making any scan, just moving the tip from one position to another fresh area and applying ramped voltage stresses at randomly selected positions, ensuring that damage due to lateral forces produced during the scans are avoided.

Figure R2 | CAFM maps collected in sequence on multilayer h-BN samples. a-f, consecutive CAFM topography and current maps collected simultaneously by using a CONTV-PT tip (with a spring constant 0.2 N/m). The maps are centred at the same position of the sample, and the topographic maps clearly indicate no damage to the surface of the sample.

Figure R3 | CAFM maps collected on CVD h-BN using a solid diamond tip applying high force. a, CAFM topography map collected by using an AD-40-AS diamond tip (with a spring constant 40 N/m). **b**, Zoom-out CAFM topography map collected at the same point where the scan in **a** was collected, using a NCHV-A Si tip (in tapping mode). It is clear that the scan with the diamond tip damaged the surface of the sample. **c**, Cross-sectional TEM image after CAFM scan in **b**; h-BN has been removed and folded at the end of the scan.

Laminating a thin Au nanoflake on the h-BN would increase the effective area (A_{eff}) across which electrons can flow (due to the high lateral conductivity of the Au film), which would prevent us from getting nanoscale resolution. This is explained in our previous CAFM book (see Lanza et al. *Conductive Atomic Force Microscopy: Applications in Nanomaterials*, 2017. Wiley-VCH, ISBN: 978-3-527-34091-0); for the reviewer's convenience, we reprint the relevant image here as Figure R4.

Figure R4 | Schematic of the effective emission area. Effective emission area through which electrons can flow (A_{eff}) in a CAFM when the tip is placed on (a) a flat insulating sample and (b) a flat metallic electrode deposited on an insulating sample.

3) The authors seemed to believe that lower V_{on} means higher quality of h-BN. Theoretically speaking, perfect crystals should have higher breakdown voltage.

We thank the reviewer for this comment. In fact, that is not what we believe, and we did not want to express such thing in our manuscript. We are sorry if our text confused the reviewer. If the reviewer can point which sentence gave him/her such impression, we could modify it.

What we think is that (for the same thickness) higher V_{ON} means higher quality of h-BN because it is more insulating. This is shown in Figure 7 of our manuscript and discussed throughout the manuscript. In the revised version of this manuscript, we have double-checked that there is no confusing sentence about this point.

The CVD grown h-BN has a much higher V_{on} . Does that mean the quality of monolayer CVD h-BN in author's lab is higher than the mechanically exfoliated one?

This is a very good point, and we thank the reviewer for highlighting it. As Figure 7 of our manuscript shows, the V_{ON} of **multilayer** mechanically exfoliated h-BN samples is always higher than in CVD h-BN samples (for the same thickness). This is consistent with the lower number of defects in the mechanically exfoliated samples, as shown via cross-sectional TEM (compare Supplementary Figure 9 with Supplementary Figures 14-18).

However, when analysing the value of V_{ON} for **monolayer** h-BN samples produced by mechanical exfoliation (Figure 2f) and CVD method (Figures 3f,l and 4e), the value for CVD h-BN samples is higher. In the case of Figures 3f,l the reason is clearly that the h-BN is not really monolayer, but it has thicknesses of ~ 2 and ~ 1.3 nm (which increases V_{ON}). However, the I-V curves shown in Figure 4e apply to a sample which seems to be indeed mainly monolayer, as seen from the cross-sectional TEM images (Figure 4a). The possible reasons for the higher V_{ON} of monolayer h-BN grown by CVD compared with mechanical exfoliated are: (i) the presence of wrinkles and multilayer islands, as seen from large-area SEM images (Figure 4b); (ii) the presence of impurities in the CVD h-BN sample from the CVD process; (iii) oxidation of the underlying Cu substrate, which oxidizes much more easily than Ru; and (iv) the presence of a slightly higher band gap in the CVD h-BN, as for the mechanically exfoliated large pressure during transfer to the Ru/Ta/SiO₂/Si substrate was applied. This explanation has been included in the revised version of the manuscript.

4) Surface roughness on Cu foil is much higher than the one on Ru film. Does it affect the Von measured? For a rough surface, the contact area between the surface and tip are relatively small.

We thank the reviewer for this interesting comment. No, the surface roughness of the Cu film does not play a relevant role. The parameter that has the highest influence is the tip-sample contact force (see Ranjan et al. *Microelectronics Reliability* 64, 172–178, 2016). In our study this was maintained always at the same value for different samples, and hence it can be ruled out.

5) In Fig. 3e, the authors attributed the conducting area to the amorphous h-BN. However, since no in-situ TEM data were performed, the authors should adopt a more conservative saying.

We thank the reviewer for this comment. Mechanically exfoliated samples show no defects in the TEM images (see Supplementary Figure 14), and the current maps don't show pinholes or low resistivity spots. This was also observed in other studies, such as Britnell, L. et al. *Nano Letters* 12, 1707-1710, 2012. On the contrary, CVD-grown samples show many defects in the TEM (see Supplementary Figures 5-6 and 8-12) and the current maps show multiple isolated current spots even if no bias is applied. This indicates that the conductive spots are related to the local defects (few-nanometre-wide amorphous regions) in the CVD-grown h-BN.

Figure R5 | CAFM characterization of mechanically exfoliated h-BN and CVD-grown h-BN. Panel **a,c** is data of mechanically exfoliated h-BN, panel **b,d** corresponds to CVD h-BN. **a**, cross-sectional TEM image of mechanically exfoliated h-BN. **b**, cross-sectional TEM image of CVD ‘monolayer’ h-BN from supplier 1. **c**, CAFM current map of mechanically exfoliated h-BN, collected without applying bias, in an area of $10 \mu\text{m} \times 10 \mu\text{m}$, shows no current. **d**, CAFM current map of CVD “monolayer” h-BN from supplier 1, collected without applying bias. Current above 100 pA can be observed.

6) In Fig. S1c, please explain the steps and noise burr observed. Besides, please use the standard resistance (for example 1 M Ohms) to determine the offset voltage and noise of narrow voltage scan.

We thank the reviewer for this comment. The stepped nature of the curve in the central part of Figure S1c is related to the minimum step of the voltage source of the Bruker Dimension Icon CAFM, which is $\sim 0.3 \text{ mV}$. For example, we can tell the voltage source to setup the voltage at 1 mV, but we cannot tell the voltage source to

set 1.215478953 mV. The voltage source has not such a high precision, it works in small steps. The noisy current signal within each step is related to small resistance fluctuations in the tip-sample system, due to by instabilities of the tip-sample contact force and electrical noise in the voltage applied by the CAFM. These two behaviours are characteristic when the CAFM tip is measuring on a metallic sample, and they have been also observed in other CAFMs, such as Multimode V (see Hui et al. *Nanoscale*, 8, 8466–8473, 2016). This explanation has been included in the caption of Supplementary Figure 2.

Answers to the comments from reviewer #2

This paper reports the benchmarking of commercially available CVD-grown hBN (both monolayer and multilayer), comparing with monolayer hBN grown in the authors' lab and multilayer hBN exfoliated from NIMS hBN. The quality assessment of commercial hBN samples is very important for many researchers, because hBN is a key insulating material for 2D research. The authors found that all the commercially available CVD-hBN products do not meet the specifications supplied from the companies.

We thank a lot to the reviewer for indicating that the topic covered by our manuscript is very important for many researchers.

I think that most of the readers are interested in which company sells the best quality hBN or in the actual quality, such as thickness, crystallinity, of the hBN samples which they are using or they plan to use. However, in this manuscript, the authors do not provide the company names (though Fig. 1 is somehow related to the company name), thus the readers cannot get enough information related to their research.

[REDACTED]

[REDACTED]

Figure R1 | Example of another study that reports on the quality of commercial 2D materials (in this case Graphene) without revealing the identity of each company. a) Graphene content per number of companies. b) Number of companies related with the number of layers from AFM (D50 and D90). Reproduced from Kauling, A. P. et al. *Advanced Materials* 30, 1803784, 2018.

[REDACTED]

Furthermore, since the authors simply examined cross-section TEM images, CAFM data, and SEM images of different samples, it lacks scientific discussion how the hBN quality comes from each CVD process. In other words, scientific insight and findings are not new enough. Therefore, I think that this manuscript does not meet the high standard of Nature Communications and, thus, cannot recommend the publication.

We are sorry that we cannot agree with this statement. The amount of TEM, SEM and CAFM data presented in our manuscript to support the conclusions is much higher than that presented in many other manuscripts in the field of 2D materials published in top Nature journals. Normally most authors present just one TEM image, and when doing CAFM only present one/few I-V curves. On the contrary, our manuscript presents many TEM images in the main text and supplementary information, and the CAFM plots contain data from hundreds of positions. We are not aware of any article showing more TEM and CAFM image about the samples, if the reviewer thinks this is not enough, we would like to kindly request to provide a few references of articles that provide more data than ours.

Regarding the discussion on the CVD process for each sample, this is something that is never included in articles of this type (see Kauling, A. P. et al. *Advanced Materials* 30, 1803784, 2018 and Bøggild, P. *Nature*, 502-503, 2018). The reason is that the companies are not willing to disclose such information. We have sent Emails to several suppliers asking for the growth temperature and this question was never answered. In fact, no article that uses commercial samples have ever disclosed such information; if the reviewer is aware of any, we would be happy if he/she could share the reference.

The followings are some small comments on this manuscript.

1. The details of the CVD growth of monolayer hBN in the authors' lab is not well described in the experimental section. Please describe the details of the CVD synthesis, such as the feedstock, CVD temperature, and Cu foil source, because their hBN is used as the standard monolayer sample.

We thank the reviewer for this constructive comment. In the revised version of the manuscript, the required information has been included in the Methods section. The new text included reads as follows:

“The CVD h-BN samples grown in our laboratory use a standard ~25- μ m-thick Cu foils from Thermo Fisher Scientific as substrate. After introduced in the tube, the temperature is raised to 1050 °C for 30 minutes in a H₂ (10-20 sccm) and Ar (5-10 sccm) atmosphere for annealing and surface cleaning. After that, the valve that controls the flow of the precursor (ammonia borane, H₃NBH₃, 95%, from Sigma-Aldrich) into the chamber is opened and the temperature is adjusted at 80 °C during the h-BN growth. Finally, the precursor valve is closed, and the temperature is ramped down to room temperature naturally. The SiO₂ samples were synthesized via thermal oxidation on n⁺⁺ Si substrates in an industrial foundry.”

2. The TEM image of their monolayer hBN (Fig. 4a,b) looks like bilayer hBN. Is this really monolayer? More details explanation is required.

We thank the reviewer for this comment, and we apologize for the confusion created. In the revised version of the manuscript, we replaced Figure 4a by a new one in which the h-BN layer can be better seen (see Figure R2).

(Continue in next page)

Figure R2 | TEM characterization of in-house CVD-grown monolayer h-BN grown in our lab. Cross-sectional high-resolution TEM image of the CVD-grown monolayer h-BN sample on Cu foil.

3. Figure caption of Fig. 6 is incorrect. “suppliers from 3-9” (from 7 companies) should be “suppliers from 4, 5, 6, 7, 9” if I can understand the explanation in the main text.

We thank the reviewer for finding this typo. In the revised version of the manuscript, we have corrected it.

4. It is better to describe why they used Ru/Ta as an electrode.

We thank the reviewer for this constructive comment. The reason that we choose Ruthenium is because it is inert to most other chemicals, not easy to form oxides, and has good electrical conductivity. Moreover, in our group we have the ability of making it very flat (root mean square roughness below 200 pm), which is necessary for CAFM studies. Tantalum is deposited below ruthenium as an adhesion layer. We have also tried Pt, and Au as an electrode, but the surface roughness we achieved is much worse than Ru, so at last we select Ru as a substrate. In the revised version of this manuscript, we have included one sentence to discuss this point.

Answers to the comments from reviewer #3

The manuscript by Yuan et al. reports the structural and electrical characterization of multilayer and “monolayer” CVD-grown h-BN film samples supplied by the main commercial suppliers, and the results are compared with mechanically exfoliated flakes based on h-BN crystals provided by the NIMS group as a reference of high-quality (standard) h-BN films. The statistical analysis presented here is of great value for the community and the conclusions of the work are clear: the properties of CVD h-BN films provided by the most popular suppliers are very far from those advertised in their product specifications, regarding film thickness, crystal quality, homogeneity and most critically, electrical properties. In general, I think the conclusions reached by Yuan et al. may serve as a starting point to bring common understanding on the actual quality of samples provided by commercial suppliers, and hopefully to speed up the process of standardization in the market of commercial 2D-materials.

We thank a lot to the reviewer for indicating that our manuscript is of great value for the community, and that the conclusions of the work are clear.

However, whether this study deserves publication on Nature Communications or other more specialized journals, I would leave this to the editor to judge, as I do not find any new science in this paper. Following are some comments the authors should address before publication, either on Nature Communications or elsewhere.

We thank the reviewer for this comment. We knew before starting to write this manuscript that CVD-grown h-BN still possesses lower quality than mechanically exfoliated. This is known by the community, as the reviewer well pointed. This kind of thinking has been extracted from device-level studies. For example, when using mechanical exfoliation, graphene transistors with h-BN substrate show mobilities of $60,000 \text{ cm}^2 \text{ V}^{-1} \text{ s}^{-1}$ (see Dean et al. Nature Nanotechnology 5, 722-736 2010), and when the same experiments are repeated using CVD h-BN

the mobility observed is only $2,500 \text{ cm}^2\text{V}^{-1}\text{s}^{-1}$ (see Pandey et al. IEEE Trans. Electron Dev. 65, 4129-4134, 2018). However, the quality of CVD-grown 2D materials evolves, and an updated benchmarking study of the available commercial suppliers is necessary.

The goal of this manuscript is to evaluate the status of commercial CVD-grown h-BN. [REDACTED].

In the following we provide answers (including additional data when needed) to the technical comments raised by the reviewer. We think the comments raised by the reviewer have been useful and have helped to enhance the quality of our article, and hence we would like again to thank the reviewer for his/her careful revision and constructive feedback.

1. The main use of h-BN in 2D research is as a dielectric layer, and it has been shown that h-BN encapsulation can drastically increase the mobility of 2D devices. I think it would be more useful to construct devices using various h-BN samples and compare the device performance.

We thank the reviewer for this constructive comment. The figures-of-merit of electronic devices depend not only on the quality of the material as grown, but also on many other factors, such as the density of defects introduced during metal evaporation, contact resistance, residues during transfer, etcetera. For example, some studies have presented transistors with maximum on-state currents up to $\sim 750 \mu\text{A}/\mu\text{m}$ using defect-rich CVD MoS_2 (see Illarionov, Yury Yu, et al. IEEE Electron Device Letters 38, 1763-1766, 2017), while others only reached $6 \mu\text{A}/\mu\text{m}$ using defect-free MoS_2 (see Mitta, Sekhar Babu, et al. 2D Materials 8, 012002, 2020). Hence, having a device with better performance does not mean the quality of the native (as-produced) materials is better.

Therefore, fabricating electronic devices is not necessary to assess the quality of as grown 2D materials and their density of native defects. In fact, it is better not to fabricate devices, as defects introduced during device fabrication would also be involved and could lead to incorrect conclusions.

2. The cross-sectional TEM analysis, while is quite useful for thickness measurement, is not typically considered an efficient method for statistical analysis, as the experiment is time consuming and expensive while each sample can only cover regions at a scale of 5-20 microns. It would be more useful if the authors can perform other statistically more efficient analysis, e.g. Raman mapping, to characterize the homogeneity of the samples.

We thank the reviewer for this constructive comment. The lateral resolution of Raman spectroscopy is very low ($\sim 1\mu\text{m}$), and therefore it only gives averaged information about the sample, and it cannot map local defects as well as TEM and CAFM does. Nevertheless, in this revision we have conducted the experiments requested. The new data are shown in Figure R1 of this letter, and they are also included in the revised version of the manuscript as Supplementary Figure 11.

Here we transferred the CVD-grown ‘monolayer’ h-BN from supplier 1 onto 300 nm SiO_2/Si substrate, and for a better comparison, we also mechanically exfoliated h-BN flakes and transferred them onto 300 nm SiO_2/Si substrate. An inhomogeneous contrast was observed in a $100 \mu\text{m} \times 100 \mu\text{m}$ Raman map of the CVD-grown ‘monolayer’ h-BN from supplier 1 (see Figure R1h), suggesting an uneven film thickness and lack of 2D crystallinity (or presence of amorphous regions). This agrees with the cross-sectional TEM images shown in Figure 3a-b, and the CAFM current map shown in Figure 3e. From 7 out of 12 Raman spectrums, a sharp E_{2g} band of h-BN at 1367.02 cm^{-1} with a full width at half maximum (FWHM) of 16 cm^{-1} was observed, while the rest 5 out of 12 Raman spectrums show no clear peak at around 1367 cm^{-1} (see Figure R1), indicating the absence of h-BN. On the contrary, repeating these experiments on exfoliated h-BN flake show stronger signal in Raman spectrum (see Figure R1b), with a narrower FWHM of 12.8 cm^{-1} . The new text included in the revised version reads as follows:

“Moreover, we transferred this “monolayer” CVD h-BN and mechanically exfoliated h-BN onto a 300 nm SiO_2/Si substrate for Raman spectroscopy characterization (see Supplementary Figure 11), and observed that: (i) the mechanically exfoliated sample shows homogeneous and strong E_{2g} band of h-BN at 1367.02 cm^{-1} with

a full width at half maximum (FWHM) of 12.8 cm^{-1} ; and (ii) the CVD h-BN from supplier 1 shows very low and inhomogeneous signal, with a FWHM of 16 cm^{-1} , plus at some locations (5 out of 12) no E_{2g} band around 1367 cm^{-1} is detected. These data demonstrate that the differences observed between the mechanically exfoliated monolayer h-BN sample (Figure 2f) and the CVD h-BN from supplier 1 (Figure 3f) are related to the presence of pinholes, atomic defects, and thickness fluctuations in the CVD h-BN from supplier 1.”

Figure R1 | Raman characterization of mechanically exfoliated h-BN and CVD-grown ‘monolayer’ h-BN from supplier 1, transferred on 300 nm SiO₂ / Si substrates. a-e, Raman and AFM maps of mechanically exfoliated h-BN flake. **a**, optical microscopy image of one mechanically exfoliated h-BN flake. **b**, Typical Raman spectrum at 7 positions marked in **a**. A sharp E_{2g} band of h-BN at 1367.02 cm^{-1} with a FWHM of 12 cm^{-1} was observed. **c**, a $45\text{ }\mu\text{m} \times 45\text{ }\mu\text{m}$ Raman map of the intensity of h-BN E_{2g} band. **d**, AFM topography map collected at the edge of the target mechanically exfoliated h-BN flake, which is the area that marked with yellow dash line in **a**. **e**, histogram distribution in **d**, the distance between two peaks is 3.2 nm , indicating the thickness of this h-BN flake. **f-h**, Raman experiments of CVD-grown ‘monolayer’ h-BN from supplier 1. **f**, photograph of $5\text{ mm} \times 8\text{ mm}$ CVD-grown ‘monolayer’ h-BN from supplier 1 on a SiO₂ (300 nm) / Si substrate after transfer. **g**, typical Raman spectrum at 12 positions indicated in **f**. **h**, a $100\text{ }\mu\text{m} \times 100\text{ }\mu\text{m}$ Raman map of the intensity of h-BN E_{2g} band.

3. In the analysis of cross-sectional TEM (for example Figure 3a-b), how can we be sure the “amorphization” is not induced by the FIB sample preparation? For example, in Figure 3b, the Cu foil also looks remarkably damaged as compared to Figure 3a. How to make sure this is the original state of the films after sample preparation? The FIB process can easily damage ultrathin films, especially if they are composed of light elements such as C or B-N. In addition, some of the samples seem to be protected only by amorphous carbon layers. It should be pointed out that even e-beam deposited carbon or thermal evaporated carbon can damage the surface structure of 2D materials. The authors need to be very careful during their sample preparation.

We thank the reviewer for this comment. We are aware that FIB could damage the structure of some samples if it is not done carefully. We have developed a very accurate protocol for the FIB process by adjusting the stage rotation degree, current and voltage of electron beam, current and voltage of ion beam. The fact that our FIB does not damage the samples in this manuscript can be observed from:

- 1 – The carbon layer on top of the h-BN (as protection layer) was deposited by using a spin coater, which does not damage to the h-BN (see Zheng et al. *Advance Materials* 2022, 34, 2104138). In our studies, we used spin-coated PMMA as the protection layer.
- 2 – The mechanically exfoliated samples are defect free because they are extracted from a high-quality crystal, and in Supplementary Figure 9 we do not observe any defects after the FIB and TEM processes. That is perfectly consistent and shows that FIB does not introduce damage or amorphization in our samples.

In the revised version of the manuscript this explanation has been further emphasized, which required us to modify the order of the Supplementary Figures.

The reason why Cu foil in Figure 3b looks ‘damaged’ as compared to Figure 3a is because of the quality compression and contrast difference in the two images. Here we provided the raw data from Figure 3a-b. The other reason is that the scale of the two images (raw data) is different, so pixel per distance is a bit different in the two images, that’s also causing the ‘unclear’ in Figure 3b.

4. More on this, the authors mention several times throughout the manuscript the possible presence of defects, especially in the “amorphous” areas, but no clear analysis is provided. Do the authors imply here that defective areas are prone to amorphization and is the cause of the observed “amorphous” areas? Seems difficult to correlate both ideas without proper structural analysis. In addition, from the cross-sectional TEM results, one can only describe e.g. the region in Figure 5j as distorted or disordered regions, because such images do not provide conclusive information about the in-plane crystallinity. So, calling it as “amorphous” may not be correct.

We thank the reviewer for this comment. Throughout the manuscript we have used the word “defect” to refer to those areas that do not show perfect layered structure in the cross-sectional TEM images, which in the current maps appear to be more conductive. Sometimes these defects consist of one/few atoms and are observed in the TEM images as a discontinuity of a layer as interstitial atoms between layers. But some others this defective bonding propagates laterally and vertically over larger areas, and that is what we call “amorphous regions”. This is mainly observed in the “monolayer” sample from supplier 1, which is the one that exhibited the worst quality.

The in-plane crystallinity in Figure 5j (now Figure 6j in the revised version of the manuscript) can be deduced from the observation of van der Waals gaps between the different layers. That, despite the obvious changes of orientation of the layers (also observed in other articles, see Adithi, K. et al. *ACS Nano* 16, 2866-2876, 2022), can be clearly observed in Figure 5j. Without 2D in-plane crystallinity, such van der Waals gaps would not form. The multilayer samples in Figure 6 do not show “amorphous” regions and in the manuscript we are not referring to them as such. We only talked about “amorphous” regions when talking about Figure 3, which we believe in that case is correct.

In the revised version of the manuscript, that sentence has been changed as: “*We analysed the first sample from supplier 1 by collecting 25 consecutive cross-sectional TEM images and observed that only 80% were 2D layered, although with a thickness ranging from 2 to 2.3 nm, and the remaining 20% contained high density of defects, mainly erratic atomic bonding (see Figure 3a-b and Supplementary Figure 8). Sometimes these defects*”

consist of one/few atoms and are observed in the TEM images as a small discontinuity of one/few layers, or as interstitial atoms between layers (see red arrows in Supplementary Figure 8). But some others this defective bonding propagates laterally and vertically over larger areas, which creates heavily disordered quasi-amorphous regions (see yellow arrows in Supplementary Figure 8). Note that these local defects are not related to amorphization produced by the focused ion beam (FIB), as when the same experiments are carried out in mechanically exfoliated samples, perfect layered structure is observed (see Supplementary Figure 9).” We have also inserted some red and yellow arrows in Supplementary Figure 8 to further clarify this point.

5. In Figure 2f, the authors mention that the I-V measurements in mechanically exfoliated monolayer h-BN show good agreement with the calculated curves for monolayer and bilayer h-BN. However, I see that the vast majority of the grey curves (experiments) correlate with the blue line, which corresponds to a trilayer model. Why is this discrepancy in the interpretation? Furthermore, in the following section (3. CVD-grown “monolayer” h-BN), the monolayer model is assumed as reference according to the abovementioned interpretation in Figure 2f. Is this whole interpretation correct and how is this affecting the results from commercial samples?

We thank the reviewer for this comment. We have further discussed the simulations with other experts in the field and finally we have decided to remove them. The reason is that these simulations were carried out years ago when key new knowledge about h-BN was still unknown, mainly simulation parameters. They are not essential for our manuscript, and they can be removed without problem.

6. The authors did not discuss the huge increase in V_{ON} for “monolayer” CVD h-BN samples from supplier 2. What is the cause?

We thank the reviewer for pointing out this comment. The reason for this increase of V_{ON} is the higher thickness of the h-BN compared to the monolayer samples. The TEM images, SEM image and CAFM topography map (Figure 3g-h, 3i and 3j, respectively), thickness fluctuation, multilayer islands, wrinkles, and particles can be observed, which cause the high variability behaviour in V_{ON} . In the revised version, the new text included reads as follows:

“However, the value of V_{ON} is very large and variable (4.20 ± 1.23 V) indicating the presence of severe thickness fluctuations in the h-BN stack.”

7. Figure 4 shows the characterization of in-house CVD-grown monolayer h-BN. However, from the TEM image I can count at least 2 layers, and seemingly up to 4 layers at some points. Despite the films seem to have better crystalline quality than those provided by commercial suppliers, the statement of “truly monolayer” is incorrect. Why in this case the V_{ON} is the lowest even if not a monolayer?

We thank the reviewer for this comment, and we apologize for the confusion created. In the revised version of the manuscript, we replaced Figure 4a by a new one in which the h-BN layer can be better seen (see Figure R2).

Figure R2 | TEM characterization of in-house CVD-grown monolayer h-BN grown in our lab. Cross-sectional high-resolution TEM image of the CVD-grown monolayer h-BN sample on Cu foil.

8. The manuscript seems to be written in a rush and lacks many important experimental details. The figure captions were not carefully written, without clear explanation of each panel, and one needs to refer to the main text in order to understand the details. Figure 6 says results from suppliers 3-9, but one has to find out from the fine print in the last column that these samples are from suppliers 5,9,4,7,6, in a very odd arrangement. This is particularly annoying.

We thank the reviewer for finding the typo in the caption of Figure 6, which has been corrected in the revised version of the manuscript. We have also added more details to the captions of all other figures. In the revised version of the manuscript, we have added many more details as requested by the reviewer.

9. Section 4 needs to be rewritten and put the information in a more coherent way. It is quite hard to read and follow. For instance, in the beginning the authors mention they want to study the variability from the multilayer samples provided by supplier 1, but one cannot know which parameter are they studying. Looking at Figure 5, one can guess is thickness, but it is confusing since those are supposed to be multilayer samples with thickness >10 nm. The text and the description of the results is incomplete and not possible to follow clearly till the end of the section, where the text reveals there is a thickness mismatch between samples. The authors should rewrite this section of the manuscript. In the same line, it is confusing that the authors describe Figure 6 before figure 5.

We thank the reviewer for this comment. In the revised version of the manuscript, we have replaced some sentences. First, we start the section by clearly stating how the quality of the samples is analysed and compared. We wrote: “*Next, we analysed the quality of multilayer CVD h-BN from suppliers 1, 4, 5, 6, 7 and 9 (the other suppliers did not offer multilayer CVD h-BN), discussing mainly surface roughness, average thickness, thickness fluctuations, number of atomic defects, observation of pinholes and electrical homogeneity (through V_{ON}).*” Second, we have changed the order of Figures 5 and 6, so that they follow the flow of the text. And third, we have added a few more explanations of the images to drive to reader into the conclusions. Those sentences read as follows: “*Next, we collected 25 consecutive cross-sectional TEM images for all the samples from supplier 1 and all (100%) of the images show traces of layered structure, although they host large amounts of local defects and lattice distortions (see Figure 6 left column and Supplementary Figures 14-18). The layered structure seems to be more accentuated in the thinnest sample (Figure 6), and for the others the interfaces with C and Cu are blurry, the layers seem to be interrupted and have bifurcated, and for the thickest samples even oblique and even almost vertical layers can be observed. This behaviour has been also observed in 2D materials grown by other academics [25].*” And also: “*However, the good correlation between thickness (observed in TEM images) and V_{ON} (observed in I-V plots) indicates that this parameter plays the most important role (even over crystallinity and density of bonding defects).*”

10. In the second paragraph of Section 4, CAFM images of samples from supplier 5 and 9 show pinholes, but in line 3, the authors claim samples from supplier 9 was pinhole free. In addition, in paragraph 3 the authors say Samples from supplier 1 are free of pinholes while Figure 5b-n clearly show the same patterns as in Figure 6. How to understand this?

We thank the reviewer for detecting this problem. We made a mistake in the text. None of the multilayer samples show current (pinholes) when scanned at 0V, as shown in Supplementary Figure 19. The current maps presented in Figures 5 and 6 have been collected while applying 0.5 V. Hence, these currents are related to tunnelling current across the weakest locations of the h-BN stack, not to the presence of pinholes. In the revised version of the manuscript the text has been modified as follows: “*From an electrical point of view, none of the multilayer samples show pinholes, as current maps collected when applying 0 V show no current spots (see Supplementary Figure 19).*” And also: “*When measuring current maps at 0.5 V, the samples from suppliers 5 and 9 show some weak spots (the quantum tunnelling current is higher at those locations), which can represent a problem if the material is used as gate dielectric in transistors, but it has been shown to be beneficial to fabricate memristors.*”

11. If the authors try to alert the community about the real quality of the commercially available samples, details about the suppliers should be provided. Otherwise, it hurts all suppliers.

[REDACTED]

[REDACTED]

Figure R3 | Example of another study that reports on the quality of commercial 2D materials (in this case Graphene) without revealing the identity of each company. a, Graphene content per number of companies. **b,** Number of companies related with the number of layers from AFM (D50 and D90). Reproduced from Kauling, A. P. et al. *Advanced Materials* 30, 1803784, 2018.

REVIEWER COMMENTS

Reviewer #1 (Remarks to the Author):

The authors have addressed part of my concerns in last round revision mainly by giving textual explanations, and the present form has some technical advances. However, after carefully read all the other reviewer's review report and the author's response, it's not difficult to figure it out that this paper, at least in the present form, still lacks of enough scientific advances. I insist on most of my last-round comments on this paper.

As we know, flatness is a very important parameter to determine whether the ultrathin insulators can be used as the dielectrics or not. To this end, the referee thinks no one faithfully believes the commercially CVD-grown h-BN, talked about in this paper, with so many wrinkles and huge roughness can serve as the dielectrics in the future. I am not surprising that such kind of rough h-BN films will have poor electrical homogeneities and more pinholes than the mechanically exfoliated h-BN with flat surface. However, if the research objectives of this paper are the winker-free CVD-grown h-BN grown on the ultra-flat substrate, such as sapphire, which has the potential for the application of gate dielectrics, it would be much more informative and meaningful.

Even though I do not find sufficient significance on evaluating the electrical homogeneity of rough CVD h-BN, I should emphasize that this paper can serve as the excellent guidance of quantitative electrical analysis on ultrathin insulators by C-AFM. If the editor insists on publishing this paper on Nature Communications, I suggest the authors to strength this part by adding more related data of C-AFM on mechanically exfoliated h-BN.

1) I am delighted to see that authors performed strict comparison experiments to avoid the possible damage caused by the tip. The authors claimed that using soft conductive tip with a low spring constant of 0.2 N/m, rather than stiff diamond tips, can reduce the physical damage during tip scanning. This is a very important information that needs to be emphasized in the main text and supporting information. The Figure R1 and R2 should be added in the SI. Besides, to make the data more reliable, continuous C-AFM maps with more times (zero bias) on the same area are needed.

2) Expect for the softness of tips, the contact forces (setting points) and scan rate also matter for reliable C-AFM maps. Please give more detailed parameters of C-AFM measurements.

3) In Figure S6b-e, with a driving voltage of 0.8 V, the conducting maps undergo obvious changes by multiple scans. However, when zero bias was applied, the damage seems to be less. Suggest the authors to do more comparison experiments on the mechanically exfoliated h-BN, and then emphasize in the main text that "C-AFM mapping with zero bias, small contact force and soft tips is verified as a faithful way to avoid physical damage on the ultrathin insulators".

4) Compared to the old version of Figure 2f, the authors have deleted several lines, please explain why?

5) Please give the raw data of ramped IV curves on mechanically exfoliated h-BN with different thicknesses, such as bilayer, trilayer and several nanometers.

6) The authors claimed that the surface roughness of the Cu film does not play a relevant role on C-AFM measurements. Is it really true? Please use the mechanically exfoliated h-BN on flat Ru film and rough Cu film to demonstrate this.

7) The referee is not satisfied with the explanation to the steps and noise burr in Figure S2c. Please use

the C-AFM calibration module with a standard resistance to determine the offset voltage and noise of narrow voltage scan of your instrument.

Reviewer #2 (Remarks to the Author):

1) Names of companies

I understand the concern. [REDACTED]. Therefore, I do not request the disclosing the company names.

2) Analysis methods

>We are not aware of any article showing more TEM and CAFM image about the samples, if the reviewer thinks this is not enough, we would like to kindly request to provide a few references of articles that provide more data than ours.

This is not the point I mentioned. What I was concerned is that both TEM and CAFM are local measurement techniques, which does not give large-area information (even though the authors present a number of TEM and CAFM images). For macroscopic analysis, I suggest measuring optical images and Raman mapping images for hBN transferred on SiO₂ substrates. Optical and Raman mapping images will give the information on the thickness uniformity as well as the crystallinity (E_{2g} FWHM) in large areas. While the authors have provided Raman maps of exfoliated flakes and commercial CVD-grown hBN films from supplier 1 in Supplementary Fig. 11, more data from the commercial samples obtained from the other suppliers are needed.

Reviewer #3 (Remarks to the Author):

I don't have further technical comments apart from one: the cross-sectional TEM images of their in-house CVD "monolayer h-BN" (Figure R2) does not show monolayer feature. The sample is multi-layer.

Answers to the comments from reviewer #1

The authors have addressed part of my concerns in last round revision mainly by giving textual explanations, and the present form has some technical advances.

We thank the reviewer for this comment, although we cannot fully agree with it. In the past revision of the manuscript, we not only changed the manuscript providing some textual explanations, but we also included a lot of new data. The amount of data that we included in the previous (first) revision was:

- Study of rough substrates (evaporated metallic film): cross-sectional TEM images of monolayer CVD-grown 2D material transferred on e-beam evaporated Pt film (see Supplementary Figure 1).
- Study of current flow through mechanically exfoliated monolayer h-BN while no bias is applied (see Supplementary Figure 5).
- Study of current flow through mechanically exfoliated monolayer h-BN in consequence scans while bias is applied (see Supplementary Figure 6).
- Study the quality of mechanically exfoliated monolayer h-BN and CVD-grown “monolayer” h-BN by using Raman (see Supplementary Figure 11).
- Repeat the cross-sectional TEM experiment for in-house CVD-grown monolayer h-BN (see Figure 4a).
- Study of the effect of CONTV-PT probe on the CAFM experiment (see Figure R2 shown in the previous response letter).
- Study of the effect of different CAFM tips (different spin constant) on the CAFM experiment: contains both CAFM data and cross-sectional TEM image (see Figure R3 shown in the previous response letter).

The amount of data that we included in this second revision is:

- Study of the quality of commercial CVD-grown h-BN by using Raman: we provided Raman spectrums and Raman mappings for all the samples from 9 suppliers.
- Study of roughness effects: compared the surface roughness of as-grown CVD h-BN on Cu foil and CVD h-BN transferred on SiO₂/Si substrate.
- Study of the effect of CAFM tip on the surface modification: we performed consequence scan at the same area for 15 times.

However, after carefully read all the other reviewer's review report and the author's response, it's not difficult to figure it out that this paper, at least in the present form, still lacks of enough scientific advances. I insist on most of my last-round comments on this paper.

We thank the reviewer for spending his/her time to evaluate our work and to recommend improvements to our manuscript. In the following we provide new data (when needed) and additional explanations to fix those concerns.

As we know, flatness is a very important parameter to determine whether the ultrathin insulators can be used as the dielectrics or not.

The surface roughness observed on the h-BN is mainly related to the surface roughness of the underlying Cu foil. This does not affect the fabrication of devices because when the material is transferred on a flat surface it will adapt to its contour. To demonstrate this, we have taken one of the roughest monolayer CVD-grown h-BN from supplier 1, and one of the roughest multilayer CVD-grown h-BN from supplier 4, and we transferred them on ultra-flat SiO₂/Si wafers. The topographic maps (see Figure R1 of this letter) indicate that the root mean square (RMS) roughness of the surface of commercial monolayer h-BN and commercial multilayer h-BN after the transfer are 0.81 nm and 3.11 nm, respectively, which are much lower than those obtained on the Cu foil. These results have been included in the Supplementary Information of the revised version of the manuscript.

Figure R1 | Surface roughness of commercial CVD-grown h-BN. **a**, Comparison of the surface roughness of a commercial CVD-grown monolayer h-BN on the Cu foil on which it was grown and after being transferred on ultra-flat SiO₂/Si wafers. **b**, Comparison of the surface roughness of a commercial CVD-grown multilayer h-BN on the Cu foil on which it was grown and after being transferred on ultra-flat SiO₂/Si wafers.

To this end, the referee thinks no one faithfully believes the commercially CVD-grown h-BN, talked about in this paper, with so many wrinkles and huge roughness can serve as the dielectrics in the future.

We agree with the reviewer, partially. We are not trying to convince anyone that today’s commercial CVD-grown h-BN is useful for being gate dielectric in transistors. The key message of our work is: “*the quality of CVD-grown h-BN is still NOT enough for being employed as gate dielectric in transistors*”, and we explain and quantify in detail how far we are, which is a very valuable information for the community.

Dielectrics have many applications. The current properties of CVD-grown h-BN might still not fulfil the requirements for being used as gate dielectric in transistors, but our group has demonstrated in several articles that its performance is very suitable for the fabrication of memristors. See for example:

- Nature 2023, 618, 57–62. <https://doi.org/10.1038/s41586-023-05973-1>
- Nature Electronics 2020, 3, 638-645. <https://doi.org/10.1038/s41928-020-00473-w>
- Nature Electronics 2018, 1, 458–465. <https://doi.org/10.1038/s41928-018-0118-9>

Hence, saying that the commercial CVD-grown h-BN is not useful for any type of electronic device is incorrect. Note that the articles cited present statistics and even report about variability and yield. Other groups have also employed commercial CVD-grown h-BN in other applications, such as: (i) radiofrequency switches [Nature Electronics 2020, 3, 479-485. <https://doi.org/10.1038/s41928-020-0416-x>], and entropy source for encryption systems [Nanoscale, 2023, 15, 9985-9992. <https://doi.org/10.1039/D3NR00030C>].

I am not surprising that such kind of rough h-BN films will have poor electrical homogeneities and more pinholes than the mechanically exfoliated h-BN with flat surface.

Rough surface of h-BN has no relationship with poor electrical homogeneity or pinholes, the reviewer is incorrect in this statement. The electrical homogeneity is related to only two things: (i) thickness fluctuations, and (ii) presence of atomic defects (dangling bonds, impurities) [Chiu, Advances in Materials Science and Engineering, 578168 (2014), <http://dx.doi.org/10.1155/2014/578168>]. The presence of pinholes is related to discontinuities in the material and/or high density of defects that forms a conductive path across the dielectric.

Surface roughness and thickness fluctuations is not the same. One material with zero thickness fluctuations and outstanding electrical properties may exhibit a very rough surface if it is placed on a very rough substrate and keep its outstanding electrical properties. The reviewer is using the concept “surface roughness” as if it would be the same to “thickness fluctuations”, and this is incorrect. In the revised version of the manuscript, we have inserted one sentence to emphasize this difference, which reads as follows:

“Surface roughness and thickness fluctuations is not the same. The electrical homogeneity is related to only two things: (i) thickness fluctuations, and (ii) presence of atomic defects (dangling bonds, impurities)^{XX}. One material with zero thickness fluctuations and outstanding electrical properties may exhibit a very rough surface if it is placed on a very rough substrate and keep its outstanding electrical properties.”

However, if the research objectives of this paper are the winker-free CVD-grown h-BN grown on the ultra-flat substrate, such as sapphire, which has the potential for the application of gate dielectrics, it would be much more informative and meaningful.

The reviewer is insisting on having us growing CVD h-BN for its use as gate dielectric in transistors, but that is not the topic of our manuscript. The objective of our work is to understand the quality of commercial 2D materials.

Understanding the quality of commercial 2D materials is of very big global interest for a vast amount of materials scientists and electronic engineers, both in academia and industry. The previous study doing similar analysis (but for liquid-phase exfoliated graphene) [Kauling et al. Advanced Materials 30, 1803784 (2018)] has been cited more than 308 times since its publication in 2018. And now, with the global push of CVD-grown 2D materials, which are being used by companies like TSMC, Intel and Samsung, the push on the samples that we analyse in our paper is even higher [Lanza et al. Advanced Materials 2022, 34, 2207843].

Even though I do not find sufficient significance on evaluating the electrical homogeneity of rough CVD h-BN, I should emphasize that this paper can serve as the excellent guidance of quantitative electrical analysis on ultrathin insulators by C-AFM. If the editor insists on publishing this paper on Nature Communications, I suggest the authors to strength this part by adding more related data of C-AFM on mechanically exfoliated h-BN.

We thank a lot to the reviewer for indicating that our manuscript can be regarded as an excellent guidance of quantitative electrical analysis on ultrathin insulators by C-AFM. We are sorry that the reviewer cannot find sufficient significance on evaluating the electrical homogeneity of commercial CVD-grown h-BN. [REDACTED]. Moreover, there is a very big global interest on understanding the quality of commercial 2D materials. The previous study doing similar analysis (but for liquid-phase exfoliated 2D materials) [Kauling et al. *Advanced Materials* 30, 1803784 (2018)] has been cited more than 308 times since its publication in 2018. And now, with the global push of CVD-grown 2D materials, which are being used by companies like TSMC, Intel and Samsung, the push on this type of samples is even higher [Lanza et al. *Advanced Materials* 2022, 34, 2207843]. [REDACTED].

1) I am delighted to see that authors performed strict comparison experiments to avoid the possible damage caused by the tip. The authors claimed that using soft conductive tip with a low spring constant of 0.2 N/m, rather than stiff diamond tips, can reduce the physical damage during tip scanning. This is a very important information that needs to be emphasized in the main text and supporting information. The Figure R1 and R2 should be added in the SI. Besides, to make the data more reliable, continuous C-AFM maps with more times (zero bias) on the same area are needed.

We thank the reviewer for indicating that the data presented in the past revision were useful.

We are confused, Figure R1 in the previous response letter has nothing to do with this comment. Did the reviewer mean Figures R2 and R3?

Regarding the last sentence of the reviewer's comment, in Figure R2 of the previous letter we included 3 consecutive scans. The reviewer is not citing any manuscript showing a longer sequence of scans, and he/she is also not specifying how many scans he/she wants us to measure. In this revision we have included data for 17 scans in the same position of the sample, as it can be seen in Figure R2 of this letter. We first scanned an area with the size of $15\ \mu\text{m} \times 15\ \mu\text{m}$, and then we zoomed-in to a $10\ \mu\text{m} \times 10\ \mu\text{m}$ area and scanned 15 times in sequence. After that, we zoomed out and scan the area with a size of $15\ \mu\text{m} \times 15\ \mu\text{m}$ again, for comparison.

(continue in next page)

Figure R2 | CAFM topography maps collected in sequence on multilayer h-BN samples. The 1st scan and the last scan (17th) are a zoom out topography maps with size of 15 $\mu\text{m} \times 15 \mu\text{m}$, while the 2nd ~ 16th are taken at the centre of the sample.

2) Expect for the softness of tips, the contact forces (setting points) and scan rate also matter for reliable C-AFM maps. Please give more detailed parameters of C-AFM measurements.

We thank the review for this comment. The deflection setpoints used in our experiments are: +1 V during engagement, 0 V during scanning. The scan rate is 1 Hz (while scanning areas of 2 $\mu\text{m} \times 2 \mu\text{m}$, 5 $\mu\text{m} \times 5 \mu\text{m}$ and 10 $\mu\text{m} \times 10 \mu\text{m}$). In the revised version of the manuscript, this information has been included in the Methods section.

3) In Figure S6b-e, with a driving voltage of 0.8 V, the conducting maps undergo obvious changes by multiple scans. However, when zero bias was applied, the damage seems to be less. Suggest the authors to do more comparison experiments on the mechanically exfoliated h-BN, and then emphasize in the main text that "C-AFM mapping with zero bias, small contact force and soft tips is verified as a faithful way to avoid physical damage on the ultrathin insulators".

The increase of quantum tunnelling over time (number of scans that stress the same area) is something normal, but the last image without voltage confirms that no permanent damage is introduced to the sample. We have added the sentence proposed by the reviewer to the main text. We believe this sentence is well supported by Figures S6 and S7.

The reviewer wrote: “*more comparison experiments on the mechanically exfoliated h-BN*”. Which experiments? What does he/she want us to do? If the reviewer does not specify a bit more, we cannot know. As mentioned, we believe this sentence is well supported by Figures S6 and S7.

4) Compared to the old version of Figure 2f, the authors have deleted several lines, please explain why?

We thank the reviewer for this comment. We have repeated the experiments (100 I-V curves in a matrix form) at several locations and left the most representative one. We noted that the lines that increase extremely fast in the former version of the manuscript might be related to pinholes or cracks.

5) Please give the raw data of ramped IV curves on mechanically exfoliated h-BN with different thicknesses, such as bilayer, trilayer and several nanometers.

This comment is confusing. The reviewer asks for the raw data of ramped IV curves on mechanically exfoliated h-BN with different thicknesses and cites bilayer and trilayer. However, our manuscript did not measure any bilayer or trilayer sample. We are very confused and don't know to which figure the reviewer is referring. In our manuscript we studied monolayer (Figure 2) and random thicknesses above 4 layers (Figure 7, blue symbols). Here we provide the raw data for such measurements (see Figure R3 of this letter). This image has been included in the revised version of the manuscript as Supplementary Figure 5.

Figure R3 | CAFM I-V curves collected on mechanically exfoliated h-BN flakes, with different thicknesses. a-f, Plots containing 100 I-V curves collected on mechanically exfoliated monolayer with thicknesses of 0.42, 1.35, 2, 3.2, 5.3 and 5.65 nm, at different random locations of their surfaces.

6) The authors claimed that the surface roughness of the Cu film does not play a relevant role on C-AFM measurements. Is it really true? Please use the mechanically exfoliated h-BN on flat Ru film and rough Cu film to demonstrate this.

We thank the reviewer for this comment. Yes, the surface roughness does not play any role, only thickness fluctuations and defects affect electrical properties, as mentioned above [see also Chiu, *Advances in Materials Science and Engineering* Article id number: 578168, 18 pages (2014), <http://dx.doi.org/10.1155/2014/578168>].

The reviewer's suggestion is not feasible. If an atomically thin mechanically exfoliated flake of h-BN is placed on a rough substrate (surface RMS roughness higher than the thickness of the flake), the flake will not be detected. Instead, what we do is to measure the electrical properties of the same h-BN, after its growth (on a rough Cu foil) and after transferred on an ultra-flat Ru substrate. As Figure R4 of this letter show, the electrical properties are the same.

Figure R4 | CAFM characterization of CVD-grown “monolayer” h-BN. **a-c**, CAFM results of as-grown CVD “monolayer” h-BN on Cu foil. **a-b**, CAFM topography and current maps collected on the surface of h-BN/Cu without applying bias. **c**, 100 I-V curves collected on the surface of as-grown CVD h-BN on Cu foil, with 100 pA current limitation. **d-f**, CAFM results of transferred CVD “monolayer” h-BN on flat Ru film. **d-e**, CAFM topography and current maps collected on the surface of h-BN/Ru without applying bias. **f**, 100 I-V curves collected on the surface of transferred CVD h-BN on Ru film, with 100 pA current limitation.

7) The referee is not satisfied with the explanation to the steps and noise burr in Figure S2c. Please use the C-AFM calibration module with a standard resistance to determine the offset voltage and noise of narrow voltage scan of your instrument.

The value of the offset voltage was already indicated in the previous version of the manuscript: “*These currents are produced by the inherent offset voltage of the CAFM [17-18], which in our machine is around 7.25 mV.*” We have used calibration module of the CAFM and received this same value.

This behaviour is typical in all CAFMs when measure this type of samples (metallic), we have observed it in all the CAFMs that we used independently of the brand, and that includes the Multimode V, Multimode VIII, Dimension Icon, Park NX-HighVac, Nanotech, Agilent 5500, CSIstruments and Omicron. Some of these data have been already published, as mentioned in the previous version of the manuscript, and cited in the text [Hui, F. et al. Moving graphene devices from lab to market: advanced graphene-coated nanoprobe. *Nanoscale* 8, 8466-8473 (2016)].

To observe the variation of the current (noise) when the offset is compensated, we scan the Ru substrate by applying different biases (0 mV, 7.3 mV, 7.1 mV, 7.2 mV, and 7.25 mV). The current map collected is shown in Figure R5 of this letter. As it can be observed, despite compensating the voltage the current is not zero and fluctuates. This behaviour is only observed when scanning metals and has no effect when analysing the insulating materials explored in this article.

(continue in next page)

Figure R5 | CAFM maps collected with applying different biases, on Ru substrate. a-b, CAFM topography map and current map, respectively. Biases change during the scanning, as marking in the current map (b).

Answers to the comments from reviewer #2

1) Names of companies

I understand the concern. [REDACTED]. Therefore, I do not request the disclosing the company names.

Thanks for understanding this.

2) Analysis methods

>We are not aware of any article showing more TEM and CAFM image about the samples, if the reviewer thinks this is not enough, we would like to kindly request to provide a few references of articles that provide more data than ours.

This is not the point I mentioned. What I was concerned is that both TEM and CAFM are local measurement techniques, which does not give large-area information (even though the authors present a number of TEM and CAFM images). For macroscopic analysis, I suggest measuring optical images and Raman mapping images for hBN transferred on SiO₂ substrates. Optical and Raman mapping images will give the information on the thickness uniformity as well as the crystallinity (E_{2g} FWHM) in large areas. While the authors have provided Raman maps of exfoliated flakes and commercial CVD-grown hBN films from supplier 1 in Supplementary Fig. 11, more data from the commercial samples obtained from the other suppliers are needed.

We thank the reviewer for clarifying. Optical images should never be used to analyse nanomaterials, as they cannot map any local defect; they can sometimes be used to qualitatively evaluate size of the material and surface roughness in very rough samples, nothing else.

Raman spectroscopy is more precise, but the diameter of the laser light is (in the best cases) of few micrometres wide. Therefore, it gives an averaged information of the locations being analysed. Anyway, as suggested by the reviewer, we have conducted Raman experiments for most of the other commercial samples (except for the multilayer h-BN from supplier 1, as we ran out of that sample during previous experiments). The results are shown in Figure R1 and R2 of this letter, and we also included them in the revised version of the manuscript as Supplementary Figures 14 and 22. The results indicate that no CVD-grown h-BN sample has a quality like the exfoliated one.

Figure R1 | Raman characterization of CVD-grown ‘monolayer’ h-BN from suppliers 2-9, transferred on 300 nm SiO₂ / Si substrates. Each Raman spectrum plot contains 12 Raman spectra collected at 12 different positions. Each Raman map is in a size of 100 μm × 100 μm, selected of the intensity of h-BN E_{2g} band. The order from **b** to **h** follows the same order as in Supplementary Figure 8.

(continue in next page)

Figure R2 | Raman characterization of CVD-grown multilayer h-BN from supplier 2, 4, 7, 9 and 12, transferred on 300 nm SiO₂ / Si substrates. Each Raman spectrum plot contains 12 Raman spectra collected at 12 different positions. Each Raman map is in a size of 100 μm × 100 μm, selected of the intensity of h-BN E_{2g} band. The order from **a** to **e** follows the same order as in Figure 5.

Answers to the comments from reviewer #3

I don't have further technical comments apart from one: the cross-sectional TEM images of their in-house CVD "monolayer h-BN" (Figure R2) does not show monolayer feature. The sample is multi-layer.

We thank the reviewer for not having any further comment.

We have modified the part related to the homemade CVD-grown h-BN indicating that it is bilayer. In fact, this interpretation fits better with the higher V_{ON} that is detected (0.87 ± 0.24 V), compared to the mechanically exfoliated monolayer (0.21 ± 0.08). Please see the new Section 4 for more details.

REVIEWER COMMENTS

Reviewer #1 (Remarks to the Author):

The authors have tried to answer my question with great efforts. My main concern is the physical damage of AFM tip to the dielectric surface with the contact mode. The message I previously got, maybe the common sense, is the contact mode of AFM will gradually destruct the sample surface, especially for the ultrathin sample. In this paper, the authors seemingly claimed that they can achieve the nearly damage-free C-AFM measurements even at multi-scan. In fact, this is the technical foundation of the work and is also the main interest I found in this paper. This is why I suggest the authors to do more experiments to figure out what kind of experimental parameters matter for achieving the damage-free C-AFM measurement. Unfortunately, the authors choose to ignore this question.

1) Figs. S6-7 can verify the zero bias is important for the nearly damage-free C-AFM measurement. Suggest the authors to add other parameters that can influence the damage-free C-AFM measurement, such as the spring constant of the tips (different tips, Fig. R2-3 in the 1st round revision) and contact force (different setting points) in the supporting information.

2) The authors have answered my question in the response letter, and claimed to add the sentences "CAFM mapping with zero bias, small contact force and soft tips is verified as a faithful way to avoid physical damage on the ultrathin insulators" in the main text. However, when I go through the main text, the related sentences are missing. Moreover, I also do not find the sentence regarding "the surface roughness and thickness fluctuation".

Reviewer #2 (Remarks to the Author):

Thank you for adding the Raman mapping images in the revised manuscript.

I understand the authors' claim that optical micrographs do not give information on local structures, such as defects. However, I still believe that optical images are helpful to understand how uniform the hBN thickness (in other words, uniformity of the number of layers) is in multilayer hBN samples.

Therefore, I still want the authors to display optical images for each multilayer sample. If the authors add optical images, I would recommend the publication.

Answers to the comments from reviewer #1

The authors have tried to answer my question with great efforts.

We thank the reviewer for taking time to re-review our manuscript.

My main concern is the physical damage of AFM tip to the dielectric surface with the contact mode. The message I previously got, maybe the common sense, is the contact mode of AFM will gradually destruct the sample surface, especially for the ultrathin sample.

This thinking is incorrect. We are not sure from where the reviewer got this thinking. The surface of a 2D material can be damaged during an AFM scan in contact mode if a high enough contact force is applied; however, if a lower contact force is employed, one can perfectly scan the surface of a 2D material without damaging it, as demonstrated in multiple studies. Here we cite a few of them:

1. Britnell, Liam, et al. "Electron tunneling through ultrathin boron nitride crystalline barriers." *Nano letters* **12.3**, 1707-1710 (2012).
2. Lee, Gwan-Hyoung, et al. "Electron tunneling through atomically flat and ultrathin hexagonal boron nitride." *Applied physics letters* **99**, 24 (2011).
3. Celano, Umberto, et al. "Progressive vs. abrupt reset behavior in conductive bridging devices: A C-AFM tomography study." *2014 IEEE International Electron Devices Meeting*, IEEE, (2014).
4. Lanza, Mario, ed. *Conductive atomic force microscopy: applications in nanomaterials*. John Wiley & Sons, (2017).
5. Lanza, Mario. "A review on resistive switching in high-k dielectrics: A nanoscale point of view using conductive atomic force microscope." *Materials* **7.3**, 2155-2182 (2014).
6. Androulidakis, Charalampos, et al. "Tunable macroscale structural superlubricity in two-layer graphene via strain engineering." *Nature communications* **11.1**, 1595 (2020).
7. Schaab, Jakob, et al. "Electrical half-wave rectification at ferroelectric domain walls." *Nature nanotechnology* **13.11**, 1028-1034 (2018).
8. Pastore Carbone, Maria Giovanna, et al. "Mosaic pattern formation in exfoliated graphene by mechanical deformation." *Nature communications* **10.1**, 1572 (2019).

If the reviewer still insists in that contact mode AFM unavoidably damages the surface of a 2D material, we would like to kindly request him/her to provide solid references. There is a very wide community of scientists doing AFM studies in 2D materials, and we don't think it is appropriate to say that all of them are wrong based simply on what the reviewer calls "common sense".

In this paper, the authors seemingly claimed that they can achieve the nearly damage-free C-AFM measurements even at multi-scan. In fact, this is the technical foundation of the work and is also the main interest I found in this paper. This is why I suggest the authors to do more experiments to figure out what kind of experimental parameters matter for achieving the damage-free C-AFM measurement. Unfortunately, the authors choose to ignore this question.

We feel quite surprised that the reviewer says that we ignored this question. That is not true. In the previous response letter, we have provided a very clear response to this point. We first scanned an area with the size of $15\ \mu\text{m} \times 15\ \mu\text{m}$, and then we zoomed-in to a $10\ \mu\text{m} \times 10\ \mu\text{m}$ area (centred at the same location) and scanned 15 times in sequence. After that, we zoomed out and scan the area with a size of $15\ \mu\text{m} \times 15\ \mu\text{m}$ again, for comparison. The 17 scans are shown in Figure R1 of this letter (see next page). The results unequivocally indicate that the sample is not damage, because all the wrinkles (which are delicate and can easily move [Langmuir 2021, 37.22, 6776-6782. <https://doi.org/10.1021/acs.langmuir.1c00862>]) appear at identical locations. This demonstrates that the sample has not been damaged. We don't think this discussion is suitable for being included in our manuscript because it is too basic and distracts the attention of the readers.

Nevertheless, as this reviewer insists, we have inserted one small paragraph in the revised version of the manuscript, and one Supplementary Figure in the Supplementary Information, as follows:

We would like to emphasize that the use of CAFM in contact mode (with the types of tips and contact forces employed in this study) does not damage the surface of the CVD-grown h-BN samples. To confirm this, we make a test experiment that consists of collecting one $15\ \mu\text{m} \times 15\ \mu\text{m}$ topographic scan in contact mode at a random location of a h-BN sample transferred on a SiO_2/Si substrate. Then, we zoom-in and scan 15 times with a size of $10\ \mu\text{m} \times 10\ \mu\text{m}$, and finally we zoom-out and scan again with a size of $15\ \mu\text{m} \times 15\ \mu\text{m}$. Despite the wrinkles in the 2D material are known to be soft regions that could be displaced if enough force is applied [25], all the images collected (see Supplementary Figure 13) are identical and clearly demonstrate that no damage or surface modification is introduced by the tip of the CAFM.

If the reviewer still insists in that the sample might be damaged, please clearly specify which experiments are proposed and we could do them. However, all our experiments clearly indicate that the sample is not damaged when scanned in contact mode (in line with a large amount of literature), and hence there is no reason to keep insisting in this point.

Figure R1 | CAFM topography maps collected in sequence on multilayer h-BN samples. The 1st scan and the last scan (17th) are a zoom out topography maps with size of $15\ \mu\text{m} \times 15\ \mu\text{m}$, while the 2nd ~ 16th are taken at the centre of the sample.

1) Figs. S6-7 can verify the zero bias is important for the nearly damage-free C-AFM measurement.

No, Figures S6-7 are not used to verify that the zero bias is important for the nearly damage-free CAFM measurement. That is not what they are showing. As explained in the previous response letter and in the previous version of the manuscript, Figure S6 is used to demonstrate that the sample has a few pinholes at the edge (Figures S6a-c) and no pinholes in the centre (Figures S6d-h), and Figure S7 is used to show that scanning at 0.8V does not produce any damage in the sample.

I suggest the authors to add other parameters that can influence the damage-free C-AFM measurement, such as the spring constant of the tips (different tips, Fig. R2-3 in the 1st round revision) and contact force (different setting points) in the supporting information.

If one uses stiffer tips (normally made of diamond) tips with higher spring constant (normally higher than 30 N/m) and uses a contact force (normally above 240 N/nm) he/she may surely damage the 2D material, as shown in other studies that employed diamond tips and high contact forces of 400 nN [Qi, Yizhou, et al. ACS applied materials & interfaces **9**.1, 1099-1106, 2017. <https://doi.org/10.1021/acsami.6b12916>]. However, in our study we only use tips with spring constant (0.2 N/m) and contact forces (deflection setpoint 0 V) that do not damage the material, as confirmed in Figure R1 of this letter. We do not see why we should investigate the damage of the mechanical 2D material when using other tips or deflection setpoints, as they do not apply to our study.

This has been explicitly mentioned in the first sentence of the aforementioned paragraph, which we copy again here for clarity:

We would like to emphasize that the use of CAFM in contact mode (with the types of tips and contact forces employed in this study) does not damage the surface of the CVD-grown h-BN samples.

2) The authors have answered my question in the response letter, and claimed to add the sentences “CAFM mapping with zero bias, small contact force and soft tips is verified as a faithful way to avoid physical damage on the ultrathin insulators” in the main text. However, when I go through the main text, the related sentences are missing. Moreover, I also do not find the sentence regarding “the surface roughness and thickness fluctuation”.

We thank the reviewer for pointing this out. Regarding the sentence “CAFM mapping with zero bias, small contact force and soft tips is verified as a faithful way to avoid physical damage on the ultrathin insulators”, we did not add that sentence identical. Instead, we wrote the following paragraph in page 3:

We collect current maps without applying any bias in the mechanically exfoliated monolayer samples, and no current is observed (Figure 2g), meaning that the sample is mainly free of pinholes. Some locations next to the edge showed some pinholes (see Supplementary Figure 6), probably due to the higher mechanical stress during peeling, but such behaviour is not representative of the entire surface of the monolayer h-BN flake: the mechanically exfoliated monolayer h-BN is mainly pinhole free (Figure 2g). When a pinhole-free region of the monolayer h-BN flake is scanned under 0.8 V, we observe the presence of some local conductive spots with typical diameters of 3.24 ± 3.21 nm, which drive maximum currents of 20.2 pA (Figure 2h). Considering that the mechanically exfoliated h-BN is free of defects (as demonstrated in several studies [21]), these local higher currents could only be explained by a small reduction of the van der Waals gap. If the same area is scanned at 0.8 V for four times, the size and the currents driven by the spots slightly increases (see Supplementary Figure 7b-e), indicating a progressive degradation of the h-BN film. We select four current spots that appear in all scans and plot the resistance in each scan for each spot (see Figure 2i). However, the damage to the h-BN stack is not very significant because subsequent scans without bias do not show remarkable currents above the noise level and no surface modification is observed (see Supplementary Figure 7f).

Regarding the sentence: “the surface roughness and thickness fluctuation”, we did not add that sentence identical. Instead, we wrote the following sentence in page 3:

When analysing the same sample with the CAFM, the RMS surface roughness appears to be high (8.96 nm, see Figure 3d), but that is related to the morphology of the Cu foil below the h-BN sheet (see Supplementary Figure 12) and it does not affect the electrical properties of the h-BN stack — the leakage current, depends on the thickness and number of defects [24].

Note that in the previous version of the manuscript, the reference 24 was already added, which is the same reference that we used in the previous response letter to answer that comment. In conclusion, all the information was included and discussed, not using the identical words written by the reviewer but our own explanations.

Answers to the comments from reviewer #2

Thank you for adding the Raman mapping images in the revised manuscript.

We thank the reviewer for taking time to re-review our manuscript. We are happy that the reviewer thinks our Raman mapping images are satisfactory.

I understand the authors' claim that optical micrographs do not give information on local structures, such as defects. However, I still believe that optical images are helpful to understand how uniform the hBN thickness (in other words, uniformity of the number of layers) is in multilayer hBN samples. Therefore, I still want the authors to display optical images for each multilayer sample. If the authors add optical images, I would recommend the publication.

We understand the reviewer's request, as that is the usual practice when working with mechanical exfoliated samples on SO₂ substrates, which have multiple random thickness fluctuations. However, such kind of optical microscope images are pointless in CVD-grown samples that are continuous and with homogeneous thickness because they only reflect the morphology of the polycrystalline Cu substrate, and no information about the h-BN is obtained. When transferred on SiO₂/Si substrates no contrast is observed, as the thickness is homogeneous, except for the cracks that sometimes happen during transfer.

Nevertheless, as this reviewer insists, we have collected the images and included them in the revised version of the manuscript, along with the following sentences:

First, we analyse the “monolayer” h-BN samples grown by CVD method on Cu foil from 9 different suppliers using an optical microscope, after being transferred on a flat SiO₂/Si (surface roughness 200 pm) for a better contrast. The images reveal that all the CVD-grown samples are continuous, and no appreciable thickness fluctuations can be observed (see Supplementary Figure 8).

Under the optical microscope all the samples look continuous and no optical contrast is observed (see Supplementary Figure 24), indicating that the thicknesses of all the CVD-grown multilayer h-BN samples (whatever it is) seems to be homogeneous throughout the each sample.

We also put the images in this letter as Figures R1 and R2, for more convenience. The reviewer will note that we provided optical microscope images for all the “monolayer” samples, but not for all the multilayer samples. The reason is that we ran out of those samples (multilayer h-BN from Supplier 1) during the other experiments. Nevertheless, it can be concluded that there is no contrast in any image indicating that they are continuous and without thickness fluctuations within each sample. In fact, we think these images do not provide any useful information for the readers, we are just including them because this reviewer insists (although we think this is pointless and distracts the readers).

(continue in next page)

Figure R1 | Optical microscope images of mechanically exfoliated h-BN and CVD-grown “monolayer” h-BN from suppliers 1-9, transferred on SiO₂/Si substrates. a, optical microscope image of mechanically exfoliated h-BN, where large thickness fluctuation can be easily observed by the contrast. **b-j**, optical microscope images of CVD-grown “monolayer” h-BN from suppliers 1-9.

Figure R2 | Optical microscope images of CVD-grown multilayer h-BN from suppliers 4, 5, 6, 7, and 9, transferred on SiO₂/Si substrates.

REVIEWER COMMENTS

Reviewer #1 (Remarks to the Author):

I am quite surprising that the authors disagree with most of my comments and are kind of reluctant to add the multi-scan C-AFM topography maps as Figure S13 during the last round revision. As I previously pointed out, I am just curious about how the authors can do the damage-free and steady C-AFM topography and ramped IV measurements on the ultrathin insulators. To me, this is a very important and meaningful topic, since, at least, my group can't do so delicate C-AFM measurements by using the tip of Bruker PIT-V2 with a spring constant of 3.0 N/m. Of course, we also can do the nearly damage-free multi-scan topography imaging with a very small contact force by using the PIT-V2 tip. However, if we want to do the steady conductance imaging, we need to use a relatively large contact force, and thus will damage the samples. More importantly, we just got very randomly distributed ramped IV curves. Those are what we got during our C-AFM measurements, which are apparently different from the authors' observations. I want to figure it out why? This is just because you used a very soft tip? If so, does a soft tip with a relatively high contact force can also damage the samples? Is the contact force for the multi-scan AFM image same with Ramped IV? I think the authors should make it clear, and the C-AFM images using different kind of tips (such as diamond, or PIT-V2) and contact forces should be added, at least in the supporting information. This part will make this paper as an excellent guidance for C-AFM on 2D materials, otherwise I can't find enough advance in this paper. When this part is appropriately addressed, I can recommend its publication.

Besides, I should emphasize that the topographic map of C-AFM is very macroscopic, not a microscopic way, to evaluate whether this technique is destructive or not. In fact, when we look close to the Figures S7b-e, the conductive areas are gradually enlarged, thus confirming the damage of C-AFM measurements, especially at a high driving voltage of 0.8 V. Of course, this kind of phenomenon can be understood that a high field strength will accelerate the destruction of the samples.

Reviewer #2 (Remarks to the Author):

Comments from reviewer #2

Thank you for adding the optical micrograph images for hBN samples.

Although the authors claim that my comment is pointless, I still think there is some meaning to show these images, because some hBN samples contains impurities and some show part of multilayer contrast. I think that it is possible to see multilayer hBN by optical microscope if the authors use a good microscope.

Anyway, I suggest the publication with the current form.

Comments from reviewer #2 on the communication between reviewer #1 and the authors

I understand the concern by reviewer #1 that the C-AFM scan may damage the hBN surface, which can influence the authors' discussion on the hBN quality assessment. We sometimes measure C-AFM in our lab, but we do not see significant damages on 2D materials. Considering that the authors pay attention on the cantilever, using soft cantilevers, and the applied force during the measurement, I think that possible damage induced by C-AFM tip is negligible in this case. The new figure, Fig. R1 (C-AFM topography maps collected in sequence on multilayer h-BN samples) also supports that almost no damage is induced by C-AFM measurements.

I have also checked comments 1) and 2), and I feel that these are not essential points of this manuscript, because the purpose of this manuscript is to address quality of the commercially available hBN samples based on scientific analyses, mainly C-AFM and TEM. Also, the authors respond to each comment with many references and data.

Overall, although I think this manuscript does not give significant impact nor scientific new achievement, this can be published assuming that the quality assessment is useful for 2D material researchers and engineers.